



# Sclerochronological evidence of pronounced seasonality from the late Pliocene of the southern North Sea Basin, and its implications

Andrew L.A. Johnson[1], Annemarie M. Valentine[2], Bernd R. Schöne[3], Melanie J. Leng[4], Stijn Goolaerts[5]

[1]*School of Built and Natural Environment, University of Derby, Derby DE22 1GB, UK*
[2]*School of Geography and Environmental Science, Nottingham Trent University, Southwell NG25 0QF, UK*
[3]*Institute of Geosciences, University of Mainz, 55128 Mainz, Germany*
[4]*National Environmental Isotope Facility, British Geological Survey, Keyworth NG12 5GG, UK*
[5]*OD Earth & History of Life and Scientific Heritage Service, Royal Belgian Institute of Natural Sciences, 1000 Brussels, Belgium*

*Correspondence to*: Andrew L.A. Johnson (a.l.a.johnson@derby.ac.uk)

**Abstract.** Oxygen isotope ($\delta^{18}$O) sclerochronology of benthic marine molluscs provides a means of reconstructing the seasonal range in seafloor temperature, subject to use of an appropriate equation relating shell $\delta^{18}$O to temperature and water $\delta^{18}$O, reasonably accurate estimation of water $\delta^{18}$O, and due consideration of growth-rate effects. Taking these factors into account, $\delta^{18}$O data from late Pliocene bivalves of the southern North Sea Basin (Belgium and the Netherlands) indicate a seasonal seafloor range at times larger than now in the area. Microgrowth-increment data from *Aequipecten opercularis*, together with the species-composition of the bivalve assemblage and aspects of preservation, suggest a setting below the summer thermocline for all but the latest material investigated. This implies a higher summer temperature at the surface than on the seafloor and consequently a greater seasonal range. A conservative (3 °C) estimate of the difference between maximum seafloor and surface temperature under circumstances of summer stratification points to seasonal surface ranges in excess of the present value (12.4 °C nearby). Using model-constrained estimates of water $\delta^{18}$O, summer surface temperature was initially in the cool temperate range (< 20 °C) and then (during the Mid-Piacenzian Warm Period; MPWP) increased into the warm temperate range (> 20 °C) before reverting to cool temperate values (in conjunction with shallowing and a loss of summer stratification). This pattern is in agreement with biotic-assemblage evidence. Winter temperature was firmly in the cool temperate range (< 10 °C) throughout, contrary to previous interpretations. Averaging of summer and winter surface temperatures for the MPWP provides a figure for mean annual sea-surface temperature that is 2–3 °C higher than the present value



(10.9 °C nearby) and in close agreement with a figure obtained by averaging alkenone- and
TEX$_{86}$-temperatures for the MPWP from the Netherlands. These proxies, however,
respectively underestimate summer temperature and overestimate winter temperature, giving
an incomplete picture of seasonality. A higher mean annual temperature than now is consistent
with the notion of global warmth in the MPWP, but a low winter temperature in the southern

North Sea Basin suggests regional reduction in oceanic heat supply, contrasting with other
interpretations of North Atlantic oceanography during the interval. Carbonate clumped isotope
($\Delta_{47}$) and biomineral unit thermometry offer means of checking the $\delta^{18}$O-based temperatures.

## 1 Introduction

The foraminiferal $\delta^{18}$O record from the deep sea indicates that the global volume of land ice

was generally lower than now during the Pliocene Epoch (Lisiecki and Raymo, 2005), and that
global mean surface temperature (GMST) was therefore generally higher. The late Pliocene
saw the last mainly-warm interval before the change to the typically cooler-than-present
conditions of the Pleistocene. During this interval, the Mid-Piacenzian Warm Period (MPWP;
3.28–3.03 Ma; Dowsett et al., 2019), 'warm peak average' temperature was 2–3°C higher than

now, similar to the GMST predicted for the end of the present century (Dowsett et al., 2013).
As evident under current global warming, the mid-Piacenzian temperature anomaly was not
uniform, being for instance greater than the global average figure at mid-latitudes in the oceans
according to results from both proxies and modelling (Lunt et al., 2010). Despite general
agreement, strong discrepancies between proxy and model estimates of mean annual sea-

surface temperature (MASST) have been identified in some regions (Dowsett et al., 2011).
Those formerly recognized in the northern North Atlantic Ocean have been reduced by limiting
proxy estimates to one source—alkenone index—and adjusting model boundary conditions
(Dowsett et al., 2019). It is, however, widely considered (e.g. Robinson, 2009; Bova et al.,
2021) that alkenone index reflects temperature in the warmer part of the year, and the same is

now thought to be generally the case for another commonly utilized geochemical proxy, the
Mg/Ca ratio of foraminiferal calcite (Bova et al., 2021). The species-composition of
assemblages of pelagic micro-organisms (particularly Foraminifera) has been extensively used
to derive both summer and winter sea-surface temperatures for the Pliocene (e.g. Dowsett et
al., 2010). The methodology, employing information on the seasonal temperatures associated

with modern representatives and relatives, assumes constancy of niche ('ecological
uniformitarianism'; Vignols et al., 2019) and, furthermore, that both summer and winter



temperatures exert an influence on modern occurrence. This is questionable for the many forms that 'bloom' in summer, and those (dinoflagellates) that survive winter as cysts (dinocysts).

It would be possible to obtain a more accurate estimate of regional MASST for comparison with model outputs by combining temperatures from a summer proxy with those from a winter proxy, if such existed. Dearing Crampton-Flood et al. (2020) obtained TEX$_{86}$ estimates about 6 °C lower than from alkenones for sea temperature during the MPWP in the Netherlands. They took the former data to reflect conditions during winter, when the source-organisms (archaea)

of the lipids concerned may have bloomed. Given that alkenones (produced by haptophyte algae) seem to reflect summer conditions, the mid-point between the two figures is probably close to MASST. However, we cannot be sure in the absence of information on the precise times during winter and summer that are represented, and for the same reason we cannot say whether the figures give an accurate picture of seasonality. In this paper we use

sclerochronology (investigation of the chemical and physical nature of accretionary mineralized skeletons) to obtain estimates of extreme summer and winter sea temperatures in individual years over a late Pliocene interval spanning the MPWP in Belgium and the Netherlands. The information substantially supplements initial sclerochronological estimates (Valentine et al., 2011) from these countries on the eastern side of the southern North Sea Basin

(SNSB), and complements a sizeable body of equivalent data relating to the early Pliocene (Zanclean) sequence of eastern England, on the western side of the SNSB (Johnson et al., 2009, 2021b; Vignols et al., 2019). The results serve to: (1) test estimates of seasonality and annual (average) temperature obtained from other proxies; (2) expand and refine the proxy evidence of temperature available for testing models of Pliocene climate; (3) provide an insight into the

controls on regional marine climate.

## 2 Sclerochronology and seasonality

The majority of sclerochronological studies of environment have been conducted on accretionary calcium carbonate skeletons, principally those of corals and molluscs in the marine realm. Trace element (Sr/Ca) profiles from shallow-water corals have been found to

mirror seasonal changes in surface temperature (e.g. DeLong et al., 2007, 2011) but no such close relationship exists in molluscs (e.g. Gillikin et al., 2005; Markulin et al., 2019). In view of the absence of corals (at least long-lived, colonial forms) from extra-tropical shallow-water environments and general inutility of trace (and minor) element data from molluscs for

reconstructing seasonal temperature variation, sclerochronological investigations of
palaeoseasonality in temperate and polar settings have been largely based on the $\delta^{18}$O of
molluscan carbonate. Pelagic belemnites supplement benthic molluscs as a provider of
information on Jurassic and Cretaceous conditions (e.g. Mettam et al., 2014) but after the
extinction of the former at the end of the Cretaceous the latter become the sole source of data
(e.g. Bice et al., 1996; Williams et al., 2010; Surge and Barrett, 2012; Johnson et al., 2009,
2017, 2019; Vignols et al., 2019; de Winter et al., 2020a, b). There is no doubt that temperature
exerts an influence on the $\delta^{18}$O of molluscan carbonate, but values are also affected by the $\delta^{18}$O
of the fluid from which the material was precipitated (usually taken to be equivalent to that of
ambient water) and by kinetic and more obscure 'vital' effects (e.g. Owen et al., 2002a, b;
Fenger et al., 2007; Garcia-March et al., 2011). At present it is possible only to constrain (not
specify) the $\delta^{18}$O of ambient water in ancient settings so, although precise, the temperatures
from $\delta^{18}$O thermometry are not necessarily accurate—i.e. they are questionable as absolute
temperatures. Nevertheless, assuming that kinetic and vital effects do not vary with season or
age, an assumption which is certainly valid for some molluscs (e.g. Fenger et al., 2007; Garcia-
March et al., 2011), and that water $\delta^{18}$O is constant, ontogenetic profiles are, at least in
principle, a true reflection of relative temperature and hence (from the difference between
summer and winter $\delta^{18}$O values) of seasonality.

Unfortunately, molluscan growth is often discontinuous, and interruptions are frequently
associated with seasonal temperature extremes (Schöne, 2008), so in such cases the shell $\delta^{18}$O
record does not fully reflect the range of temperatures experienced (e.g. Hickson et al., 2000;
Peharda et al., 2019a). However, because of their typical manifestation as 'growth lines',
interruptions can be recognized and instances of likely truncation of the $\delta^{18}$O record inferred
(e.g. Johnson et al., 2017, 2019, 2021b). The increasing occurrence and/or duration of growth
interruptions with age is part of the reason for the commonly observed reduction in amplitude
of $\delta^{18}$O profiles through ontogeny, but general slowing of growth (and consequent greater time-
averaging within samples) is also contributory (Goodwin et al., 2003; Ivany et al., 2003).
Increasing sample resolution can potentially offset this effect (Schöne, 2008) but the most
accurate indication of seasonal temperature variation is likely to be obtained from early
ontogenetic data (Goodwin et al. 2003; Ivany et al., 2003).


A problem as important as growth cessation/reduction for inference of seasonal temperature
range is the choice of equation relating temperature to water and shell $\delta^{18}$O. Various equations





exist for both aragonite and calcite, and for the same range in shell $\delta^{18}O$ these yield different temperature ranges. Thus for a water value of 0.0 ‰ and summer and winter shell values of

0.0 ‰ and +2.0 ‰, respectively, the widely employed aragonite equation of Grossman and Ku (1986) yields seasonal temperatures of 19.4 °C and 10.7 °C—i.e. a seasonal range of 8.7°C. However, for the same $\delta^{18}O$ values the *Glycymeris glycymeris*-specific aragonite equation of Royer et al. (2013) yields seasonal temperatures of 17.4 °C and 12.1 °C—i.e. a seasonal range of only 5.3 °C. Since both equations are linear, like the LL (low-light) calcite equation of Bemis

et al. (1998), used by Johnson et al. (2021b), neither the absolute values specifying a summer-winter difference of 2.0 ‰ in shell $\delta^{18}O$, nor the value of water $\delta^{18}O$, affect the calculated seasonal temperature range. However, the non-linear calcite equations of O'Neil et al. (1969; as reformulated by Shackleton et al., 1974) and Kim and O'Neil (1997) not only yield different temperature ranges for a given range in shell $\delta^{18}O$, but the temperature range in each case varies

with the absolute shell values concerned, and with water $\delta^{18}O$. Thus for a water value of 0.0 ‰ and summer and winter shell values of 0.0 ‰ and + 2.0‰, respectively, the calcite equation of O'Neil et al. (1969) yields a seasonal temperature range of 8.2 °C (summer 15.7 °C, winter 7.5 °C) and that of Kim and O'Neil (1997) a seasonal temperature range of 8.9 °C (summer 13.7°C, winter 4.8 °C), but for a water value of +0.4 ‰ and summer and winter shell values of +1.5 ‰

and +3.5 ‰ (i.e. the same 2.0 ‰ range but at higher absolute values), the equation of O'Neil et al. (1969) yields a seasonal temperature range of 7.8 °C (summer 11.1 °C, winter 3.3 °C) and that of Kim and O'Neil (1997) a seasonal temperature range of 8.6 °C (summer 8.8 °C, winter 0.2 °C). The differences in seasonal temperature range due to different water $\delta^{18}O$ and absolute shell $\delta^{18}O$ values are not great but the differences due to different equations are fairly

significant for calcite (up to almost 1 °C for the water and shell $\delta^{18}O$ values specified above) and quite major for aragonite (over 3 °C). Clearly, therefore, the choice of equation must be given careful consideration.

Modelling (e.g. Williams et al., 2009) and carbonate clumped isotope ($\Delta_{47}$) analysis (e.g. Briard

et al. 2020; Caldarescu et al., 2021) are techniques that have been used to constrain water $\delta^{18}O$. The studies cited in relation to the latter approach employ it to resolve seasonal fluctuations, and de Winter et al. (2021) discuss the best sampling strategy to achieve this end. In nearshore settings affected by major seasonal influxes of freshwater (normally isotopically light), and which exhibit concomitant reductions in salinity, variation in water $\delta^{18}O$ may be quite high.

Lloyd (1964) documented change of more than 1 ‰ over a few months in part of Florida Bay and Ivany et al. (2004) inferred seasonal variation of 2.5 ‰ in an Eocene nearshore setting in



the south-eastern USA. However, in more offshore settings the effects of freshwater influx are much less. Thus in the modern North Sea seasonal variation in salinity is in most places only 0.25 PSU (Howarth et al., 1993), which translates to a seasonal variation in water $\delta^{18}O$ of just

0.07 ‰ using the salinity–water $\delta^{18}O$ relationship for the North Sea of Harwood et al. (2008). Within a few tens of kilometres of the mouth of the Rhine seasonal variation in salinity rises to 0.75 PSU and hence calculated variation in water $\delta^{18}O$ to 0.21 ‰. At 20–30 m depth in the eastern part of the central North Sea, Schöne and Fiebig (2009) identified variation in salinity of up to 2 PSU in certain years, which translates to a variation in water $\delta^{18}O$ of 0.55 ‰. If

minimum and maximum water $\delta^{18}O$ values differing by this amount coincided respectively with the times of maximum and minimum water temperature it would increase the temperature range calculated from shell $\delta^{18}O$ assuming constant water $\delta^{18}O$ by an amount in the order of 2.6 °C (figure for calcite using the LL equation mentioned above). However, the data of Schöne and Fiebig (2009) provide no evidence of a negative correlation between salinity/water $\delta^{18}O$

and temperature, and near the eastern shore of the central North Sea there is a very strong positive correlation between water $\delta^{18}O$ and temperature over the seasonal cycle (Ullmann et al., 2011; de Winter et al., 2021). This presumably reflects relatively high evaporation in summer, combined with relatively low freshwater input, a common pattern in mid-latitude settings and one suggesting that seasonal variation in shell $\delta^{18}O$ of marine organisms at mid-

latitudes is more likely to be damped than enhanced by variation in water $\delta^{18}O$.

Even if there were a negative correlation between water $\delta^{18}O$ and temperature it would be confined to nearshore waters (more susceptible to freshwater influx), hence the effect on calculated temperature range could be mitigated by use of offshore shells. This approach

introduces the possibility of underestimation of the surface range as a result of life positions below the summer thermocline (typically at 25–30 m depth in shelf settings). However, shells from sub-thermocline settings may be recognized from the associated sediments and biota, and, in the case of the scallop *Aequipecten opercularis*, from microgrowth-increment patterns (Johnson et al., 2009, 2021b; Fig. 1).




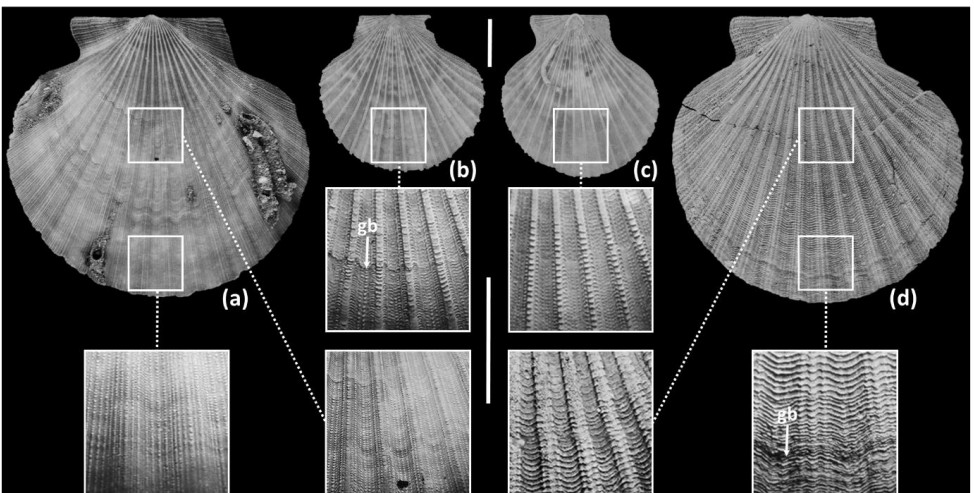

**Figure 1:** Pictorial demonstration of microgrowth-increment patterns in left valves of *Aequipecten opercularis*. **(a)** Typical supra-thermocline specimen (mesotidal setting, 23 m depth, La Coruña, Galicia, Spain) showing small increments early and late in ontogeny. **(b, c)** Typical sub-thermocline specimens (b: microtidal setting, 50 m depth, Gulf of Tunis, Tunisia; c: microtidal setting, 38 m depth, Adriatic Sea, Pula, Croatia) showing large increments early in ontogeny. **(d)** Inferred sub-thermocline specimen (Ramsholt Member, Coralline Crag Formation, Broom Pit, Suffolk, UK) showing large increments early in ontogeny and a transition from large to small increments late in ontogeny. Scale bars for whole-shell images (upper) and enlargements (lower) = 10 mm. Major growth breaks (gb) identified in enlargements of (b, d). (a) = University of Derby, Geological Collections (UD) 53424; (b) = National Museum of Natural History, Paris, IM-2008-1542 (one of seven specimens in this lot); (c) = UD 53423 (one of 48 specimens in this lot, coded S3A29); (d) = UD 53425. See Johnson et al. (2009, 2021b) for numerical data and discussion of microgrowth-increment patterns in *A. opercularis*. Modern supra-thermocline specimens show a difference of < 0.3 mm between the maximum and minimum values of smoothed increment-height profiles, while the majority of sub-thermocline specimens show a difference of > 0.3 mm.

While $\delta^{18}O$ sclerochronology is potentially informative about seasonality, it should be clear from the foregoing that results from the technique need to be interpreted carefully. The reliability of the information is of course also dependent on preservation of the original shell $\delta^{18}O$ signature.

### 3 Setting and material

In the Pliocene the marine area of the SNSB was somewhat greater than now (Fig. 2), partly due to higher global sea-level and partly to subsequent regional uplift (Westaway et al., 2001). To the west, onshore marine deposits exist in eastern England, and to the east in Belgium and the Netherlands, those in the last two countries passing eastwards into essentially fluvial non-marine deposits of the proto-Rhine/Meuse/Scheldt river system (Louwye et al., 2020; Munsterman et al., 2020). The Eridanos river system, draining the Baltic area, had its exit into





the SNSB in the area of the present German Bight, some 400 km north-east of the proto-Rhine/Meuse/Scheldt exit (Gibbard and Lewin, 2016). While at certain times a link may have

existed between the SNSB and the Channel Basin during the Pliocene (either at the present position or across southern England; Funnell, 1996; Westaway et al., 2002; van Vliet-Lanoë et al., 2002; Gibbard and Lewin, 2016), at others the basins were separated by the Weald–Artois land-bridge, as shown in Fig. 2. Water depth in the southern North Sea is now less than 40 m in most places but seismic stratigraphy indicates that it was greater in the Pliocene, at least in

areas of low sediment accumulation (Overeem et al., 2001).

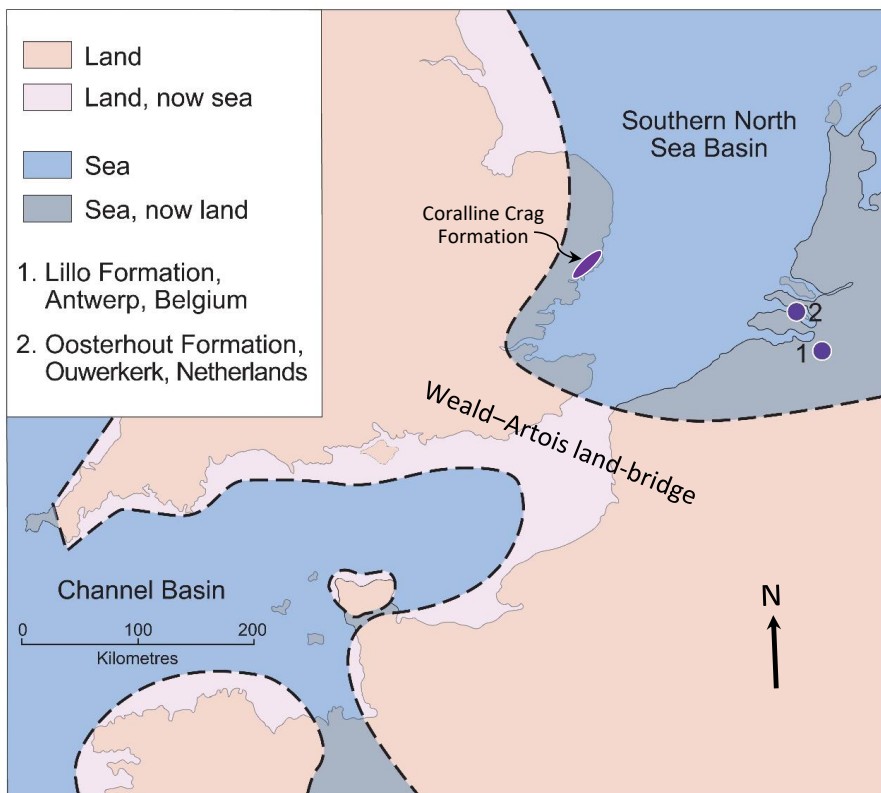

**Figure 2:** Pliocene palaeogeography in the vicinity of the SNSB, the location of sites in the Lillo (1) and Oosterhout (2) formations from which shells were obtained, and the area of onshore outcrop of the Coralline Crag Formation in eastern England (the partly Pleistocene Red Crag Formation occurs over a larger area). Adapted from Valentine et al. (2011, fig. 1), itself based on Murray (1992, map NG1).


In eastern England there is a large stratigraphic gap between the Zanclean Coralline Crag Formation and the late Piacenzian basal unit ('Walton Crag') of the Red Crag Formation, but the Piacenzian is better recorded in northern Belgium by the Lillo Formation and in the south-



west Netherlands by the Oosterhout Formation (Fig. 3). The last two formations essentially

comprise marine sands, the Oosterhout Formation at Ouwerkerk (Zeeland) probably deposited in deeper water than the Lillo Formation at Antwerp from the evidence of fish otoliths (Gaemers and Schwarzhans, 1973) and a position farther from the inferred shoreline (Fig. 2). Nevertheless, Slupik et al. (2007) inferred a depth of deposition above storm wavebase for the Oosterhout Formation at Schelphoek, 15 km north-west of Ouwerkerk. In the Antwerp area

depth estimates based on the fauna have varied between authors according to the group studied, most of them hardly taking into account the marked variation in sediment and sedimentary structures within members of the Lillo Formation (see Deckers et al., 2020, figs. 4–6). According to Gaemers (1975), the otolith assemblage indicates a depth of at least 10–20 m for the 'Kallo Sands' (= Lillo Formation, Oorderen Member; Marquet and Herman, 2009) but less

than 10 m for the overlying Kruisschans Member. This indication of upward shallowing is supported by assemblage evidence from dinocysts (Louwye et al., 2004; De Schepper et al., 2009), Foraminifera (Laga, 1972) and bivalves (Marquet, 2004), but statistical data from the last group suggest greater absolute depths: 35–45 m for the Oorderen Member and 15–55 m for the Kruisschans Member by the common overlap in depth-range of extant species; 40–50

m for the former and 20-50 m for the latter by the medial depth of extant species. The articulated preservation of the semi-infaunal bivalve *Atrina fragilis*, locally in life position, within the Oorderen Member (Marquet and Herman, 2009) is difficult to reconcile with the 10–20 m minimum depth estimate of Gaemers (1975), since specimens would have been subject to fair-weather processes after death. It is more likely that they lived at the depth

suggested by Marquet (2004) and were killed by rapid burial (and permanently interred) in storms. A somewhat greater depth still was inferred from the bivalve assemblage of the underlying Luchtbal Member: 40–50 m by 'common overlap'; 40–60 m by 'medial depth'. The low diversity of the bivalve fauna of the Merksem Member (overlying the Kruisschans Member) precluded the same statistical treatment but Marquet and Herman (2009) inferred

from this impoverishment a depth of less than 15 m, an estimate consistent with the foraminiferal assemblage (Laga, 1972) and the high proportion of terrestrial palynomorphs (De Schepper et al., 2009).

Dinocyst assemblages indicate surface temperatures within the warm temperate range (but

possibly only with respect to summer; Sect. 1) during deposition of most of the Oorderen Member, but punctuated by cool intervals and preceded by continuously cool conditions during deposition of the Luchtbal Member (De Schepper et al., 2009). The dinocysts of the





Kruisschans and Merksem members mainly indicate a continuation of the warm conditions of
the Oorderen Member but provide a few hints of cooling (Louwye et al. 2004; De Schepper et

al., 2009). Other evidence of this is provided by bivalves, fish and pollen (Hacquaert, 1961;
Vandenberghe et al., 2000; Marquet, 2005), and Wood et al. (1993) determined a 5–6 °C
decrease in summer surface temperature from the ostracods of a contemporaneous part of the
Oosterhout Formation.

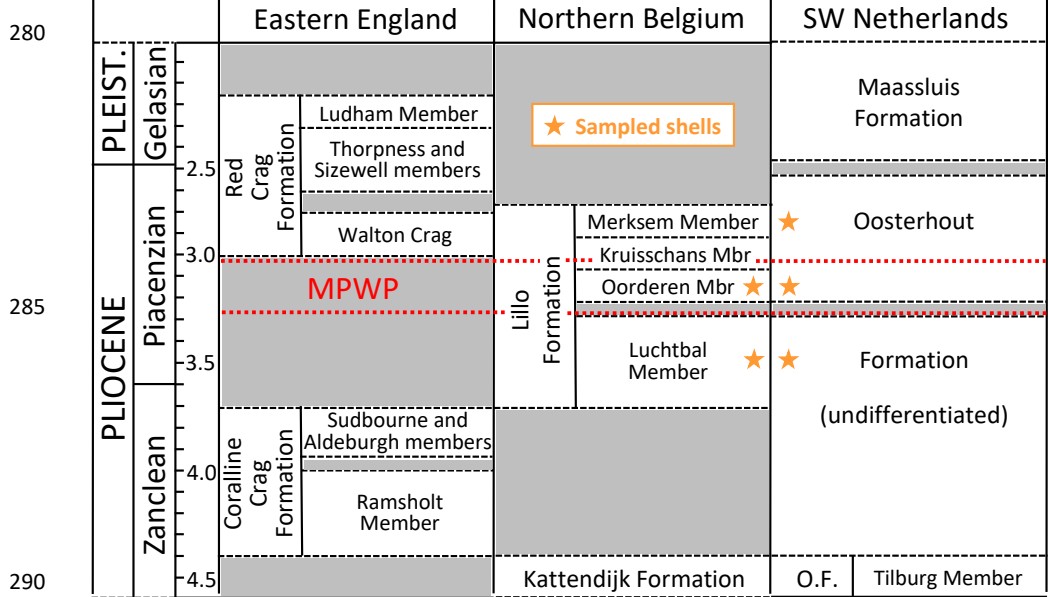

**Figure 3:** Stratigraphy and correlation of marine mid–late Pliocene and early Pleistocene units of the
southern North Sea Basin, with the general stratigraphic positions of shells sampled for the present
study (specific positions in Table 1). Age (Ma) of the Red Crag Formation and constituent members
(including the unofficial Walton Crag unit) according to Wood et al. (2009); of the Coralline Crag,

Kattendijk and Lillo formations and constituent members according to De Schepper et al. (2009); and
of the Oosterhout (O.F.) and Maassluis formations according to Dearing Crampton-Flood et al. (2020)
and Wesselingh et al. (2020). An additional small hiatus, of uncertain age, is present in the lower part
of the Oosterhout Formation (Dearing Crampton-Flood et al., 2020). The Maassluis Formation includes
a number of non-marine horizons (Slupik et al., 2007). Names of Lillo Formation members are in

accordance with recent practice (Louwye et al., 2020; Wesselingh et al., 2020), omitting
'Sands'/'Sands', as included by previous authors. Wesselingh et al. (2020) found evidence of an
additional layer (Broechem Unit) between the Kattendijk Formation and Luchtbal Member of the Lillo
Formation. De Meuter and Laga (1976) designated an additional, uppermost division of the latter
formation (Zandvliet Member), but this may be no more than the decalcified top of the Merksem

Member (Louwye et al., 2020). Geographic provenance of shells and the location of the Coralline Crag
Formation shown in Fig. 2. MPWP = Mid-Piacenzian Warm Period.





**Table 1:** Basic information for the investigated specimens (all single valves). Order stratigraphic within formations (by member, then by borehole-depth or bed, with entries for the Oosterhout Formation inserted immediately above those, if any, for the equivalent member in the Lillo Formation). Entries in square brackets are interpretations (see footnotes). Latitudes and longitudes are for the location indicated in the adjacent column and do not necessarily specify the exact place of collection (see Fig. 2 for the positions of Ouwerkerk and Antwerp). UD = University of Derby, Geological Collections; IRSNB = Royal Belgian Institute of Natural Sciences, Brussels (MSNB was used for specimens discussed in Valentine et al., 2011).

| Formation | Member or equivalent (in quotes) | Borehole-depth (b-d) or bed | Location | Latitude, longitude | Genus and species | Repository and number | Code herein | Valve height (mm) | General physical condition | Mineralogy of sampled layer | Number of isotope samples |
|---|---|---|---|---|---|---|---|---|---|---|---|
| Oosterhout | 'Merksem' | b-d: 89.75–91 m | Ouwerkerk | 51.626° N, 3.983° E | *Aequipecten opercularis* | UD 53362 | AO10 | 56 | Incomplete | Calcite | 42 |
| Oosterhout | 'Merksem' | b-d: 93.5–94.5 m | Ouwerkerk | 51.626° N, 3.983° E | *Aequipecten opercularis* | UD 53363 | AO9 | 46 | Incomplete | Calcite | 30 |
| Oosterhout | 'Oorderen' | b-d: 98.5–99.5 m | Ouwerkerk | 51.626° N, 3.983° E | *Aequipecten opercularis* | UD 53347 | AO8 | 34 | Incomplete, abraded | Calcite | 31 |
| Lillo | Oorderen | *Atrina fragilis* bed | Vrasenedok, Kallo, Antwerp | 51.263° N, 4.238° E | *Aequipecten opercularis* | IRSNB Invert-29710-10 | AO7 | 42 | Complete | Calcite | 31 |
| Lillo | Oorderen | *Atrina fragilis* bed | Vrasenedok, Kallo, Antwerp | 51.263° N, 4.238° E | *Aequipecten opercularis* | IRSNB Invert-29710-09 | AO6 | 51 | Complete | Calcite | 39 |
| Lillo | Oorderen | Base *Atrina fragilis* bed | Deurganckdok, Doel, Antwerp | 51.291° N, 4.257° E | *Aequipecten opercularis* | IRSNB Invert-D2-8 | AO5 | 31 | Complete | Calcite | 23 |
| Lillo | Oorderen | *Atrina fragilis* bed | Vrasenedok, Kallo, Antwerp | 51.263° N, 4.238° E | *Pygocardia rustica* | IRSNB Invert-29710-04 | PR | 62 | Complete | Aragonite | 37 |
| Lillo | Oorderen | *Atrina fragilis* bed | [Antwerp]ᵃ | 51.217 ° N 4.421° E | *Arctica islandica* | IRSNB Invert-18201-01 | AI | 64 | Complete | Aragonite | 32 |
| Oosterhout | 'Luchtbal' | b-d: 106–107.5 m | Ouwerkerk | 51.626° N, 3.983° E | *Aequipecten opercularis* | UD 53364 | AO4 | 44 | Incomplete, abraded | Calcite | 36 |
| Lillo | Luchtbal | *Palliolum gerardi* bed | Deurganckdok, Doel, Antwerp | 51.291° N, 4.257° E | *Aequipecten opercularis* | IRSNB Invert-29710-13 | AO3 | 42 | Complete | Calcite | 28 |
| Lillo | Luchtbal | *Palliolum gerardi* bed | Deurganckdok, Doel, Antwerp | 51.291° N, 4.257° E | *Aequipecten opercularis* | IRSNB Invert-29710-12 | AO2 | 47 | Complete | Calcite | 30 |
| Lillo | Luchtbal | *Palliolum gerardi* bed | Deurganckdok, Doel, Antwerp | 51.291° N, 4.257° E | *Aequipecten opercularis* | IRSNB Invert-29710-11 | AO1 | 54 | Complete | Calcite | 28 |
| Lillo | Luchtbal | [lower bed]ᵇ | Deurganckdok, Doel, Antwerp | 51.291° N, 4.257° E | *Glycymeris radiolyrata* | IRSNB 7698 | GR2 | 77 | Broken in storage | Aragonite | 74 |
| Lillo | Luchtbal | [lower bed]ᵇ | Deurganckdok, Doel, Antwerp | 51.291° N, 4.257° E | *Glycymeris radiolyrata* | IRSNB Invert-29710-0062 | GR1 | 92 | Broken in storage | Aragonite | 42 |

a    no specific location indicated within Belgium, but highly likely to be Antwerp
b    species indicated as 'special' to the lower bed of the Luchtbal Member in the Deurganckdok (Marquet, 2002)

Previous sclerochronological investigation of late Pliocene temperatures in Belgium and the Netherlands focussed on the Oorderen Member and an equivalent horizon in the Oosterhout Formation, and was restricted to $\delta^{18}$O data from two bivalve species, *Aequipecten opercularis* and *Atrina fragilis* (Valentine et al., 2011). Here we supplement the existing $\delta^{18}$O data from *A. opercularis* with microgrowth-increment data from the same specimens to gain an insight into their hydrographic setting (sub- or supra-thermocline) and also supply $\delta^{18}$O data from two further bivalve species (*Arctica islandica* and *Pygocardia rustica*) from the Oorderen Member, and another (*Glycymeris radiolyrata*) from the Luchtbal Member. In addition, we provide *A. opercularis* data from the Luchtbal Member and horizons equivalent to the Luchtbal and Merksem members in the Oosterhout Formation. Values for $\delta^{13}$C (obtained alongside $\delta^{18}$O) are reported for all species. Details of the provenance of the specimens are given in Table 1, together with alphanumeric codes (AO = *A. opercularis*; AI = *A. islandica*; PR = *P. rustica*; GR = *G. radiolyrata*) and sundry basic descriptive information. Note that all the specimens





from the Oorderen Member come from the *Atrina fragilis* bed, a horizon with the 'warm'
dinoflagellate biota found at most levels in the member. Illustrations of species other than *A.*

*opercularis* (Fig. 1) are provided in Fig. 4. Most, if not all, of the material from the Lillo
Formation was obtained from temporary exposures created during harbour works in the
Antwerp area, while all the material from the Oosterhout Formation was obtained from a
borehole (Rijkswaterstaat-Deltadienst, afdeling Waterhuishouding, 42H19-4/42H0039) at
Ouwerkerk, Zeeland. Interpretation of positions (depths) within the Ouwerkerk borehole in

terms of members within the Lillo Formation follows Gaemers and Schwarzhans (1973) except
in the case of AO8, for which we have accepted the opinion of F. Wesselingh (in Valentine et
al., 2011) that the position is equivalent to the Oorderen Member. Gaemers and Schwarzans
(1973) considered that strata of this age ('Kallo Sands') were missing at Ouwerkerk but they
appear to be well represented at Schelphoek, only 15 km away (Slupik et al., 2007).

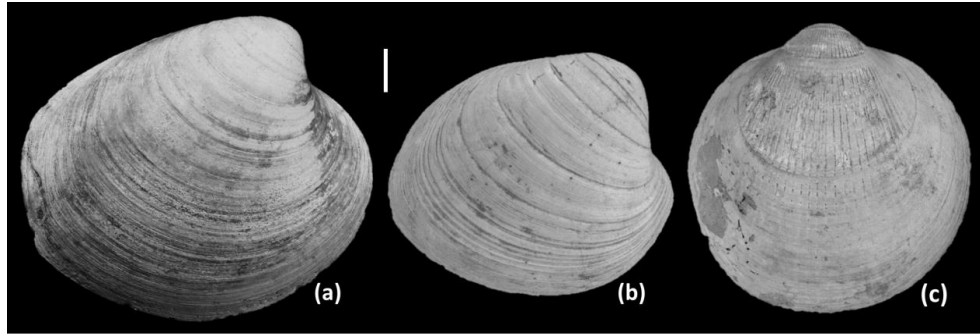


**Figure 4:** Right valves of **(a)** *Arctica islandica*, **(b)** *Pygocardia rustica* and **(c)** *Glycymeris radiolyrata*
from the Lillo Formation, Antwerp. (a): probably Oorderen Member, Verrebroekdok (IRSNB 7699);
(b): Oorderen Member, Verrebroekdok (IRSNB 7700); (c): Luchtbal Member, Deurganckdok (IRSNB
7701). Growth lines/breaks are evident in all three specimens—e.g. *c.* 10 major growth breaks in (b).

Scale bar = 10 mm.

According to the latest chronostratigraphy (Fig. 3), the material investigated is largely or
entirely Piacenzian (3.60–2.59 Ma) in age, the oldest (from the Luchtbal Member of the Lillo
Formation) being possibly as old as 3.71 Ma (latest Zanclean) and the youngest (from horizons

in the Oosterhout Formation equivalent to the Merksem Member of the Lillo Formation) being
no younger than 2.76 Ma (De Schepper et al., 2009). The MPWP is probably represented by
material from the Oorderen Member and the equivalent level in the Oosterhout Formation
(Valentine et al., 2011). All the specimens come from stratigraphic intervals with a fully marine
associated biota (e.g. Marquet, 2002, 2005; Gaemers and Schwarzhans, 1973), in conformity

with modern occurrences in the case of the extant species *A. opercularis* and *A. islandica*





(Tebble, 1976) and other fossil occurrences in the case of the extinct species *P. rustica* and *G. radiolyrata* (Norton, 1975; Buchardt and Simonarson, 2003; Marquet, 2002, 2005). Investigation of specimens of *A. islandica*, *P. rustica* and *G. radiolyrata* added information from infaunal, slow-growing taxa to that derived from fast-growing, epifaunal *A. opercularis*,

hence serving to mitigate any 'ecological' bias in the results. We could not sample as many specimens of the infaunal, slow-growing species as of *A. opercularis* due to the limited availability of material (perforce from museums, in the lack of extant stratal exposures in the area of study). However, we sampled multiple years in the infaunal, slow-growing species so the combined number of seasonal cycles investigated was similar to that in *A. opercularis*. We

nevertheless expected some imbalance in the data because modern examples of *Glycymeris* species, from both cool- and warm-temperate settings, show winter cessation or slowing of growth and thus supply (or would supply) underestimates of the seasonal temperature range from $\delta^{18}O$ sclerochronology (Peharda et al., 2012, 2019a, b; Royer et al., 2013; Reynolds et al., 2017; Featherstone et al., 2020; Alexandroff et al., 2021). Various equations have been used to

express the precise relationship between $\delta^{18}O$ and temperature in modern *Glycymeris* (Royer et al., 2013; Peharda et al., 2019a, b) but species of this genus certainly exhibit something at least close to equilibrium isotopic incorporation. The same is true of *A. opercularis* (Hickson et al., 1999; Johnson et al., 2021b) and *A. islandica* (Schöne, 2013; Mette et al., 2018; Trofimova et al., 2018). Because of the similarity of $\delta^{18}O$ values from seemingly well-

preserved *P. rustica* from the Pliocene of Iceland to those from co-occurring, similarly preserved *A. islandica* (Buchardt and Simonarson, 2003) it is reasonable to assume equilibrium fractionation in the former (extinct) species. The specimens analysed showed no obvious signs of alteration and they are unlikely to have suffered significant heating through burial as the thickness of overlying sediments was probably never much more than 100 m (the depth below

the present surface of the lowermost shell from the Ouwerkerk borehole). Examples of both calcitic *A. opercularis* and aragonitic *A. fragilis* from the Lillo Formation exhibit the original shell microstructure (Valentine et al., 2011), as do examples of a variety of calcitic and aragonitic species from the slightly earlier Ramsholt Member of the Coralline Crag Formation in eastern England (Johnson et al., 2009; Vignols et al., 2019). We therefore considered it

reasonable to proceed with isotopic analysis of our material (both calcitic and aragonitic; see Sect. 4 and Table 1) without detailed investigation of its preservation.

**4 Methods**

### 4.1 Laboratory procedures

The exterior of *A. opercularis* shells was coated with a sublimate of ammonium chloride and digitally photographed for the purpose of measuring microgrowth increments and the position of growth breaks. The coating was washed off with tap-water and the shells then underwent the further cleaning procedure adopted by Valentine et al. (2011) for removal of any surficial organic matter, in preparation for isotopic sampling of the outer shell layer from the exterior, as in other such investigations of *A. opercularis* (e.g. Hickson et al., 1999, 2000; Johnson et al., 2009, 2021b; Vignols et al., 2019). The infaunal species were sampled in cross-section along the line of maximum growth, in accordance with universal practice for *A. islandica* (e.g. Schöne et al., 2005) and common practice for *Glycymeris* species (e.g. Royer et al., 2013). For this purpose shells were stabilized in resin before sectioning, by the partial-encasement method of Schöne et al. (2005) for *A. islandica* and *P. rustica,* and the total-encasement method of Johnson et al. (2021a) for *G. radiolyrata* (fragments bonded beforehand). Use of vacuum impregnation in the latter method resulted in resin penetration into the outer part of the outer shell layer.

Extraction of isotope samples from *A. opercularis* shells was by drilling a dorsal to ventral series of shallow commarginal grooves (depth and width < 1 mm; cf. Hickson et al., 1999, fig. 2; 2000, fig. 3) in the external surface, with the sample sites more closely spaced towards the ventral margin in an attempt to maintain temporal resolution in the face of declining growth rate with age. Details of the procedure are given in Johnson et al. (2019) with respect to another scallop species. Mean sample spacing for individuals—the average distance between the centres of grooves along the dorso-ventral (= maximum-growth/shell-height) axis—was 0.93 (AO8)–1.35 (AO9) mm. Sampling of the infaunal species was by drilling a series of holes (depth and width < 1 mm; Fig. 5) in the outer shell layer as seen in cross-section, the curved path being located about midway between the external surface and the boundary between the outer and inner shell layers in *A. islandica* and *P. rustica*, but somewhat closer to the latter boundary in *G. radiolyrata* to avoid resin-contaminated material (Fig. 5). Sample spacing was more constant than for *A. opercularis*, although significantly reduced late in the long series from *G. radiolyrata* GR2, again to maintain temporal resolution. Mean sample spacing for individuals—the average distance between the centres of holes, measured in terms of the difference in straight-line distance from the origin of growth—was 0.69 mm for *A. islandica*, 0.57 mm for *P. rustica*, and 0.54 (GR2) and 0.57 (GR1) mm for *G. radiolyrata.* Note that in



these relatively convex species the straight-line distance from the origin of growth is not a measurement of shell height as normally defined (a distance from the umbo, which protrudes dorsal of the origin of growth in these forms; e.g. Fig. 5) and that the plane in which it was measured (along the line of maximum growth) arguably does not include the shell height axis

in the prosogyrate species *A. islandica* and *P. rustica* (dependent on the point at the shell margin that is regarded as ventral). The lines of measurement and the values obtained are, however, regarded as 'heights' for all four species considered herein, for the sake of simplicity. The *A. opercularis* shells were relatively small (Table 1) and were sampled from near the origin of growth (dorsal margin) to a point at or close to the ventral margin (maximum sample height

53.0 mm in AO10). The shells of the infaunal species were larger (Table 1) and not sampled to the end of ontogeny (maximum sample height 54.7 mm in GR2). Furthermore, the thinness of the outer layer close to the origin of growth meant that sampling had to start relatively far from this point (minimum sample height 15.4 mm in GR2).

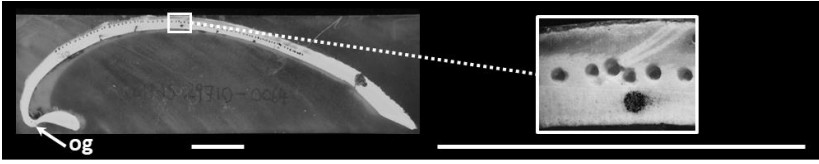

**Figure 5:** Cross-section of *Glycymeris radiolyrata* specimen GR2 showing the origin of growth (og), position of sample holes (relatively far from the external surface in this species to avoid the darker, resin-contaminated material) and a major growth break (pale diagonal band in enlargement) at shell height 35.4 mm. Scale bars = 10 mm. Black spot in enlargement is a marker to assist sample numbering.

The cross-sections of the infaunal species were digitally photographed for the purpose of

measuring the positions of sample holes and growth breaks, as seen on the shell exterior (cf. Fig. 4) and projected or traced (Fig. 5) into the isotope sample path. Distances from the origin of growth were determined from the images using the bespoke measuring software Panopea© (2004, Peinl and Schöne). Panopea was also used to measure the position of growth breaks and the height of microgrowth increments in the shell-exterior images of *A. opercularis* (cf. Fig.

1). As in the case of isotope sample positions, measurements were made along the dorso-ventral axis or (where this was impossible due to abrasion or encrustation) lateral to this line, the measurements then being mathematically adjusted as described by Johnson et al. (2019) to correspond to ones made along the dorso-ventral axis. All the microgrowth-increment measurements were made by the same person (AMV), thus assuring a reasonably uniform

approach given the subjective element in increment identification (Johnson et al., 2021b).



Growth breaks were classified as major (incorporating 'moderate') or minor in all species dependent on their external prominence (cf. Figs. 1, 4).

Samples (typically (50–100 µg) were analysed for their stable oxygen and carbon isotope
composition (given as $\delta^{18}O$ and $\delta^{13}C$) at the stable isotope facility, British Geological Survey, Keyworth, UK (*A. opercularis*, *A. islandica*, *P. rustica*) and the Institute of Geosciences, University of Mainz, Germany (*G. radiolyrata*). At Keyworth, samples were analysed using an Isoprime dual inlet mass spectrometer coupled to a Multiprep system; powder samples were dissolved with concentrated phosphoric acid in borosilicate Wheaton vials at 90°C. At Mainz,
samples were analysed using a Thermo Finnigan MAT 253 continuous flow–isotope ratio mass spectrometer coupled to a Gasbench II; powder samples were dissolved with water-free phosphoric acid in helium-flushed borosilicate exetainers at 72°C. Both laboratories calculated $\delta^{13}C$ and $\delta^{18}O$ against VPDB and calibrated data against NBS-19 (preferred values: +1.95 ‰ for $\delta^{13}C$, –2.20 ‰ for $\delta^{18}O$) and their own Carrara Marble standard
(Keyworth: +2.00 ‰ for $\delta^{13}C$, –1.73 ‰ for $\delta^{18}O$; Mainz: +2.01 ‰ for $\delta^{13}C$, –1.91 ‰ for $\delta^{18}O$). Values were consistently within ± 0.05 ‰ of the values for $\delta^{18}O$ and $\delta^{13}C$ in NBS-19. Note that $\delta^{18}O$ of shell aragonite was not corrected for different acid-fractionation factors of aragonite and calcite (for further explanation see Füllenbach et al., 2015).

### 4.2 Calculation of temperatures

In previous work on late Pliocene bivalves from Belgium and the Netherlands, minimum and maximum estimates of global average seawater $\delta^{18}O$ (–0.5 ‰ and –0.2 ‰), and minimum and maximum modelled values for the early Pliocene in the western part of the SNSB (+0.1 ‰ and +0.5 ‰), all adjusted downwards by 0.1 ‰ to allow for the input of isotopically light freshwater into the eastern SNSB, were used to calculate sets of temperatures from shell $\delta^{18}O$ (Valentine
et al., 2011). It seems appropriate to apply the adjusted modelled values more widely to late Pliocene material from Belgium and the Netherlands, but the adjusted global values are probably unreasonably low and hence not worth applying here (Johnson et al., 2021b).

Valentine et al. (2011) employed the calcite equation of O'Neil et al. (1969) for calculation of
temperatures from *A. opercularis* but there are grounds for thinking that this provides slightly inaccurate figures (Hickson et al., 1999; Vignols et al. 2019). The LL calcite equation of Bemis et al. (1998) seems to provide more accurate figures and certainly yields a larger estimate for seasonal range (Johnson et al., 2021b). Both equations have therefore been employed herein to





generate 'minimum' and 'maximum' seasonal ranges from *A. opercularis*. Note that the calcite
equation of Kim and O'Neil (1997) yields an intermediate estimate but the absolute
temperatures obtained from *A. opercularis* are too low (Johnson et al., 2021b).

Just as there is some uncertainty as to the best equation for calculation of temperatures from *A.
opercularis* calcite, so different equations have been favoured for use with aragonitic
*Glycymeris glycymeris*. Royer at al. (2013) advocated use of a species-specific equation
developed by them, while Reynolds et al. (2017) provide grounds for using the general
aragonite equation of Grossman and Ku (1986). The former yields a smaller estimate of
seasonal range than the latter so again both have been employed herein in relation to *G.
radiolyrata.* The equation of Grossman and Ku (1986) is generally used in relation to aragonitic
*A. islandica* and supplies similar temperatures from co-occurring (also aragonitic) *P. rustica*
specimens (Buchardt and Simonarson, 2003). This, and no other equation, has therefore been
used herein in relation to these species. In calculating temperatures appropriate adjustments
were made to allow for the different scales used in measurement of water (VSMOW) and shell
(VPDB) $\delta^{18}$O values (Coplen et al., 1983; Vignols et al., 2019).




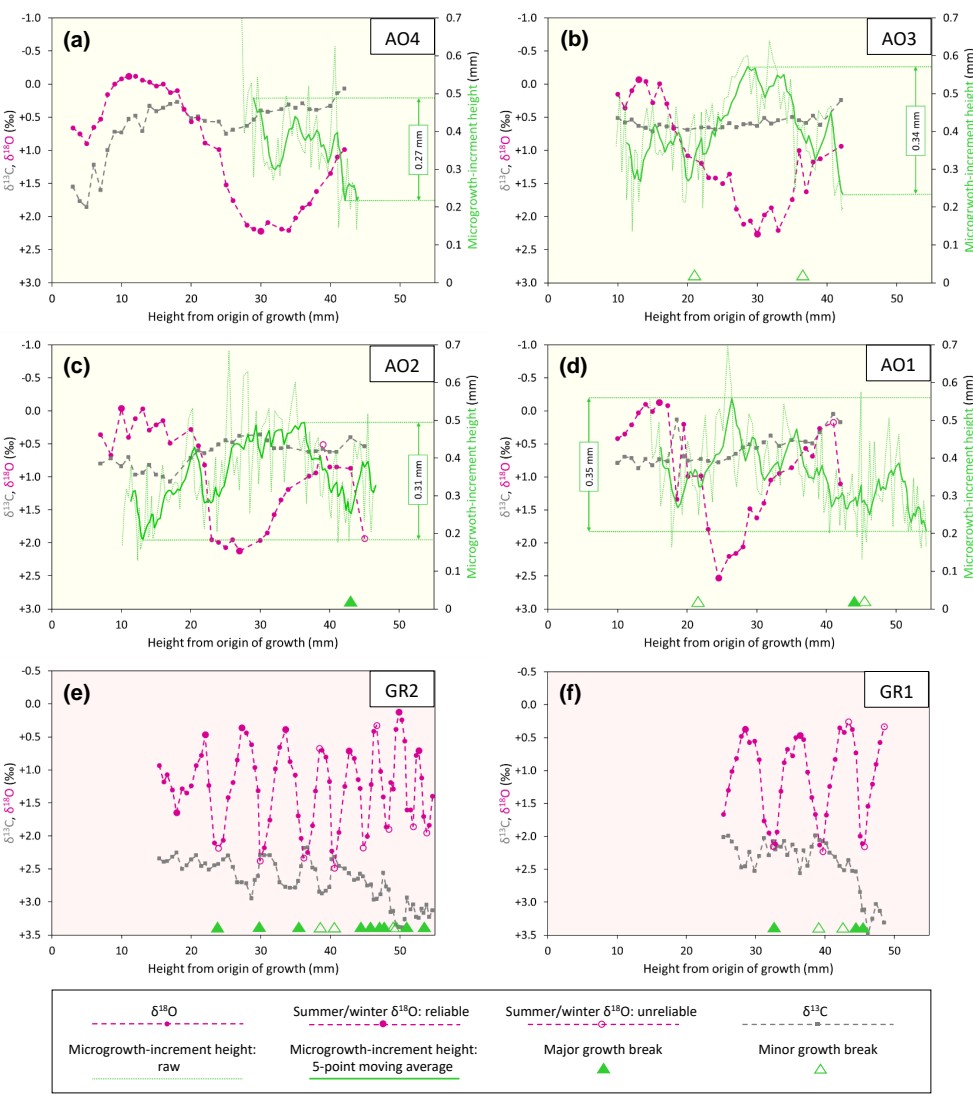

**Figure 6:** Ontogenetic profiles of $\delta^{18}O$, $\delta^{13}C$ and microgrowth-increment height from Luchtbal Member (and equivalent) *A. opercularis* **(a–d)** and *G. radiolyrata* **(e, f)**. Note that the isotopic axis has been reversed in each part such that lower values of $\delta^{18}O$ (corresponding to higher temperatures) plot towards the top. While the axis range is 4‰ throughout, the minimum and maximum values for *A. opercularis* (calcitic; pale yellow background) have been set 0.5‰ lower than for *G. radiolyrata* (aragonitic; pale pink background) to facilitate comparison, given the different fractionation factors applying for $\delta^{18}O$ (Kim et al., 2007). The criteria for recognition of reliable and unreliable summer and winter $\delta^{18}O$ values are given in Sect. 6.1.1. The fairly large single-point $\delta^{18}O$ excursion at height 18.5 mm in (d) can be regarded as 'noise' (like smaller excursions in other $\delta^{18}O$ profiles) because it is associated with an aberrant $\delta^{13}C$ value.

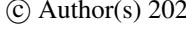



**5 Basic results and analysis**

The isotopic, microgrowth-increment and growth-break data are shown in Figs. 6 (Luchtbal

Member and equivalent) and 7 (Oorderen Member and equivalent; Merksem-Member
equivalent). Read top to bottom, left to right (i.e. in the alphabetical order of parts), the
sequence in each figure is as in Table 1, read top to bottom. The raw data is available online
(Johnson et al., 2021c).

**5.1 δ¹⁸O values and growth breaks**

Disregarding 'noise' (e.g. Fig. 6c), all profiles show cyclical patterns of $\delta^{18}O$ variation, from
less than half a cycle in *A. opercularis* profiles starting near the origin of growth and
terminating at a height of about 30 mm (AO8, AO5—Fig. 7c, f, respectively) to between two
and three in a profile terminating at 53 mm (AO10—Fig. 7a), but from between two and three
cycles to substantially more over smaller height intervals in *G. radiolyrata* (between three and

four from 25–49 mm in GR1—Fig. 6f; between eight and nine from 15–55 mm in GR2—Fig.
6e), and in *P. rustica* and *A. islandica* (between two and three from 27–48 mm in each case—
Fig. 7g, h, respectively). In *A. opercularis* profiles extending beyond one $\delta^{18}O$ cycle the
amplitude commonly shows a clear ontogenetic decrease. This pattern is less pervasive and
pronounced amongst the other species, and the *A. islandica* specimen shows an ontogenetic

increase in amplitude. However, the lack of early ontogenetic data for comparison from these
species should be noted. The maximum amplitudes from *G. radiolyrata* specimens are less than
from most *A. opercularis* specimens but those from *P. rustica* and *A. islandica* are similar to
*A. opercularis*. Growth breaks, albeit sometimes only minor, are associated with (< 1 mm from
the sample sites of) nearly all $\delta^{18}O$ maxima and a few $\delta^{18}O$ minima from *G. radiolyrata*, but

with none of the maxima or minima from *A. opercularis*. Growth breaks are associated with
two of the three maxima and two of the three minima from the *A. islandica* specimen, and with
two of the three minima from the *P. rustica* specimen.

Taking the $\delta^{18}O$ cycles to reflect seasonal temperature variation and hence intervals of a year,

the much smaller number over a given height interval from *A. opercularis* confirms that this
species grew a great deal faster than the others. In *A. opercularis* profiles spanning two or more
years (AO10, AO6 —Fig. 7a, e, respectively) there is an ontogenetic decrease in wavelength
as well as amplitude—i.e. growth was fastest in early ontogeny. This pattern has been widely
documented in *A. opercularis* from both $\delta^{18}O$ and other evidence (e.g. Johnson et al., 2021b),



and in the present instances (in which $\delta^{18}$O maxima and minima are not associated with growth breaks) the ontogenetic decrease in amplitude of $\delta^{18}$O cycles is probably a consequence of the general slowing of growth with age, leading to time-averaging in samples. Whatever the explanation, seasonal temperature variation is likely to be most faithfully reflected by the first $\delta^{18}$O cycle in *A. opercularis* profiles. The profiles from *G. radiolyrata*, *P. rustica* and *A.*

*islandica* undoubtedly omit several early ontogenetic cycles and given the short wavelength of the later cycles represented it may be that the amplitude of these is reduced by time-averaging, as inferred in *A. opercularis*. Even if the closer spacing of samples from *G. radiolyrata*, *P. rustica* and *A. islandica* may have been sufficient in principle for resolution of seasonal $\delta^{18}$O extremes, the association of growth breaks with maxima, minima or both suggests that some

recorded extremes are not representative of the most extreme temperatures experienced by the organism in the season concerned—i.e. $\delta^{18}$O variation may not fully reflect seasonal temperature variation.



**Figure 7:** Isotopic, microgrowth-increment and growth-break data from Merksem-equivalent *A. opercularis* (**a, b**) and Oorderen Member (and equivalent) *A. opercularis* (**c–f**), *P. rustica* (**g**) and *A. islandica* (**h**). Format and symbols as in Fig. 6.





### 5.2 δ¹³C values

Compared to $\delta^{18}O$ values from the same specimen, $\delta^{13}C$ values show less variation, usually substantially so. While in some specimens there are intervals of ontogeny exhibiting covariation between $\delta^{13}C$ and $\delta^{18}O$ (moderate positive covariation in AO10, AO6, PR—Fig. 7a, e, g, respectively; moderate–strong negative covariation in GR2, GR1—Fig. 6e, f, respectively), the general picture is of fluctuations (if any) in $\delta^{13}C$ that are independent of $\delta^{18}O$.

The *A. opercularis* specimens show a marginal to clear overall decrease in $\delta^{13}C$ through ontogeny, while the *P. rustica* specimen shows little change and the *A. islandica* and *G. radiolyrata* specimens show clear overall increases. The mean values from the *A. opercularis* specimens are very similar—from $+0.31 \pm 0.22$ ‰ ($\pm 1\sigma$) in AO8 to $+0.77 \pm 0.24$ ‰ in AO9— and comparable to the mean from the *P. rustica* specimen ($+0.98 \pm 0.18$ ‰), but much lower

than the means from the *A. islandica* ($+2.44 \pm 0.35$ ‰) and *G. radiolyrata* (GR1, $+2.42 \pm 0.40$ ‰; GR2, $+2.69 \pm 0.32$ ‰) specimens. The data from *A. opercularis* and *A. islandica* compare closely with those from early Pliocene examples of these species from eastern England (Johnson et al., 2009; Vignols et al., 2019). The difference between the means from early Pliocene *A. opercularis* (calcitic) and *A. islandica* (aragonitic) was ascribed principally to the

mineralogical difference (Vignols et al., 2019). This interpretation is supported by the mean values from the present *G. radiolyrata* (aragonitic) specimens, which are similar to those from *A. islandica*, but not by the *P. rustica* (also aragonitic) mean value, which is only a little outside the range of mean values from *A. opercularis*. The different pattern of overall ontogenetic change in *G. radiolyrata* and *A. islandica* (increase, unlike in *A. opercularis* and *P. rustica*)

also remains to be explained, as does the unusual negative covariation between $\delta^{13}C$ and $\delta^{18}O$ in *G. radiolyrata*.

### 5.3 Microgrowth-increment patterns (*A. opercularis*)

Even in smoothed (5-point moving average) profiles of microgrowth-increment size from *A. opercularis*, substantial high-frequency variation is present in nearly all cases. However,

amongst those profiles long enough to show a low-frequency pattern, in a number of cases a fairly clear and complete major cycle proceeding from small to large to small increments is discernible over about the first 40 mm of shell height. Such a cycle is evident in three of the four Luchtbal-Member (and equivalent) profiles, in each case with an amplitude (difference between the maximum and minimum of the smoothed profile) of more than 0.30 mm. The

exception (AO4—Fig. 5a) is a profile too short to show this pattern. Only one (AO6—Fig. 6g)





of the four Oorderen-Member (and equivalent) profiles has an amplitude greater than 0.30 mm, but a second (AO5—Fig. 6f) has an amplitude only fractionally less and a third (AO8—Fig. 6c) is too short to show equivalent ('high amplitude') variation. Despite their considerable length the Merksem-equivalent profiles exhibit an amplitude well below 0.30 mm ('low

amplitude'). The prevalent high-amplitude pattern from Luchtbal-Member (and equivalent) shells corresponds to that in modern sub-thermocline shells, and the occurrence of the pattern in an Oorderen-Member shell is at least inconsistent with a supra-thermocline setting (Johnson et al., 2009, 2021b). The low-amplitude pattern in the two Merksem-equivalent shells is consistent with a supra-thermocline setting, but given the occasional occurrence of such a

pattern in sub-thermocline shells, is not inconsistent with the latter setting.

## 6 Interpretation

### 6.1 Temperatures

#### 6.1.1 Derivation, comparison and evaluation of seasonal seafloor values

The equations and water $\delta^{18}O$ values that were employed to calculate summer and winter

temperatures from shell $\delta^{18}O$ are explained in Sect. 4.2. Following the reasoning of Johnson et al. (2017), the shell $\delta^{18}O$ values used were the extreme ones at inflection points in profiles, supplemented in the present case by those profile-end values possibly corresponding to inflection points on the evidence of similar (true) inflection-point values from the same and other co-occurring shells of the species concerned. The profile-end values probably provide

slight underestimates of summer temperature and slight overestimates of winter temperature in some cases—i.e. had the profiles extended farther, slightly lower or higher $\delta^{18}O$ values, respectively, might have been identified. As well as errors from this source, others of the same type no doubt exist in the case of data from locations close to growth breaks (as a result of an incomplete record) and, for at least *A. opercularis*, in the case of late ontogenetic data (as a

result of time-averaging), although no significant error is likely where the temperatures concerned are higher (for summers) or lower (for winters) than a reliable estimate from another time in ontogeny. Alongside the profile-end and near-growth break (unreliable) winter $\delta^{18}O$ data from *G. radiolyrata* is one inflection-point value (the first, unaccompanied by a growth break, at a height of 17.9 mm in GR2; Fig. 6e) which appears acceptable as a source of reliable

temperature information. The $\delta^{18}O$ value is, however, lower than any from this taxon regarded as an unreliable source of winter temperatures. Hence, it too must be regarded as suspect,





possibly recording a downward temperature fluctuation in spring rather than true winter conditions.

All the calculated temperatures are represented in Fig. 8, those based on equations providing 'minimum' seasonal ranges for *G. radiolyrata* and *A. opercularis* combined in Fig. 8a, those providing 'maximum' seasonal ranges for these species combined in Fig. 8b, and those providing a 'minimum' seasonal range for *G. radiolyrata* and a 'maximum' for *A. opercularis* (a 'hybrid' set) combined in Fig. 8c. Unreliable values (probable underestimates for summers

and overestimates for winters) are identified by use of an open symbol, whereby we have applied the above reasoning uniformly except to *G. radiolyrata*, *A. islandica* and *P. rustica*: in the absence of early ontogenetic data for comparison in these species, we have assumed that the late ontogenetic data represented are free from time-averaging effects.

Figure 8 shows a general similarity in seasonal temperatures within and between stratigraphic members, with the exception of winter values supplied by *G. radiolyrata* from the Luchtbal Member, which are markedly higher than those from Luchtbal-Member *A. opercularis*. Since the specimens of each species come from different (immediately adjacent) beds it is conceivable that the data reflect environmental change. However, given the fact that the

summer temperatures supplied by the species are close or identical (dependent on the method of calculation) and that all of the 12 sets of winter temperatures from *G. radiolyrata* are probable overestimates, it seems much more likely that the change is apparent rather than real. If this is accepted then it is sensible to view those Luchtbal-Member (and equivalent) summer temperatures represented in Fig. 8a and Fig. 8c (very similar from *G. radiolyrata* and *A.*

*opercularis* for the same water δ18O value) as more accurate than those in Fig. 8b (somewhat lower from *A. opercularis* than from *G. radiolyrata* for the same water δ18O value). The Oorderen-Member winter temperatures from *A. islandica* and *P. rustica* are closer to those from Oorderen-Member *A. opercularis* (from the same bed) in Fig. 8a than those in Fig. 8c (and Fig. 8b), suggesting that the data in Fig. 8a are the most accurate overall. However, if it is

untrue that the winter data from *A. islandica* and *P. rustica* are free from time-averaging effects (i.e. if the data are unreliable) there is no reason for favouring the remaining data in Fig. 8a over those in Fig. 8c.

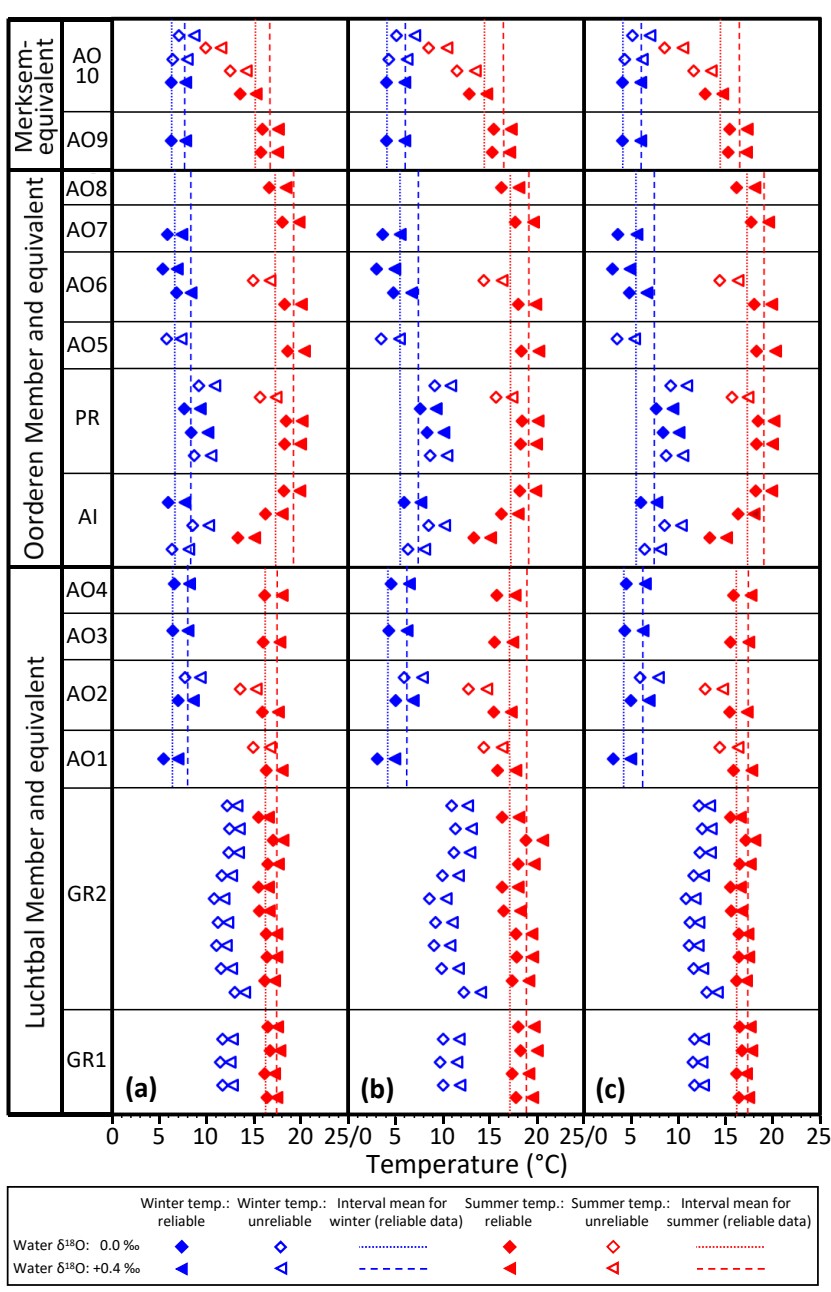


**Figure 8:** Winter and summer temperatures calculated from the seasonal extreme δ¹⁸O values indicated in Figs. 6 and 7, using water δ¹⁸O values of 0.0 ‰ and +0.4 ‰, and various equations. **(a)**: equation of Royer et al. (2013) for GR, of O'Neil et al. (1969) for AO, and of Grossman and Ku (1986) for AI and PR; **(b)**: equation of Grossman and Ku (1986) for GR, AI and PR, and of Bemis et al. (1998) for AO; **(c)**: equation of Royer et al. (2013) for GR, of Bemis et al. (1998) for AO, and of Grossman and Ku (1986) for AI and PR. Interval means are for reliable seasonal temperatures (see Sect. 6.1.1) from the Luchtbal Member and equivalent, Oorderen Member and equivalent, and Merksem-equivalent strata.




**Table 2:** Mean seasonal seafloor temperatures (°C; ± 1σ) and seafloor temperature range (summer minus winter) for 'members', based on the reliable data indicated in Fig. 8.

|  | Member and/or equivalent | Water $\delta^{18}O$ = 0.0 ‰ | | | Water $\delta^{18}O$ = +0.4 ‰ | | |
|---|---|---|---|---|---|---|---|
|  |  | Summer | Winter | Range | Summer | Winter | Range |
| Fig. 8a | Merksem | 15.1 ± 1.1 | 6.2 ± 0.0 | 8.9 | 16.8 ± 1.1 | 7.8 ± 0.0 | 9.0 |
|  | Oorderen | 17.3 ± 1.6 | 6.7 ± 1.0 | 10.6 | 19.1 ± 1.6 | 8.3 ± 1.1 | 10.8 |
|  | Luchtbal | 16.2 ± 0.4 | 6.4 ± 0.6 | 9.8 | 17.5 ± 0.5 | 8.0 ± 0.6 | 9.5 |
| Fig. 8b | Merksem | 14.5 ± 1.2 | 4.1 ± 0.0 | 10.4 | 16.4 ± 1.2 | 6.0 ± 0.0 | 10.4 |
|  | Oorderen | 17.2 ± 1.6 | 5.6 ± 2.0 | 11.6 | 19.0 ± 1.6 | 7.4 ± 1.9 | 11.6 |
|  | Luchtbal | 17.1 ± 1.1 | 4.2 ± 0.7 | 12.9 | 18.9 ± 1.0 | 6.2 ± 0.7 | 12.7 |
| Fig. 8c | Merksem | 14.5 ± 1.2 | 4.1 ± 0.0 | 10.4 | 16.4 ± 1.2 | 6.0 ± 0.0 | 10.4 |
|  | Oorderen | 17.2 ± 1.6 | 5.6 ± 2.0 | 11.6 | 19.0 ± 1.6 | 7.4 ± 1.9 | 11.6 |
|  | Luchtbal | 16.1 ± 0.5 | 4.2 ± 0.7 | 11.9 | 17.4 ± 0.4 | 6.2 ± 0.7 | 11.2 |


The interval mean values for each member (and its equivalent) shown in Fig. 8 are listed in Table 2. Unlike other changes, the Luchtbal to Oorderen increases and the Oorderen to Merksem decreases in mean summer temperature evident in Fig. 8a and Fig. 8c are statistically significant for both water $\delta^{18}O$ values (one-tailed $t$-tests; $p < 0.05$). The fact that these changes

correspond at least qualitatively to the changes in summer temperature inferred from dinoflagellates and ostracods (Sect. 3) cements the impression that the data in Fig. 8a and Fig. 8c provide a more accurate picture of seasonal temperatures, and hence of seasonal range, than the data in Fig. 8b, in which only the Oorderen to Merksem decrease in summer temperature (again statistically significant) is evident.

**Table 3:** Summer (SSFT) and winter (WSFT) seafloor temperatures (°C; precise to one decimal place; d.p.) for the year showing the greatest range (SFR) from each shell, together with an estimate of the sea-surface range (SSR). SFR calculated from non-rounded SSFT and WSFT values and then rounded (one d.p.). SSR = SFR for Merksem-equivalent (M.) shells; SSR = SFR + 3 °C for Luchtbal and Oorderen (and equivalent) shells (see Sect. 6.1.3 for explanation).

| Member and/or equiv. | Shell | Temperatures using the equations of O'Neil et al. (1969) for AO, Royer et al. (2013) for GR, Grossman and Ku (1986) for AI and PR | | | | | | | | Temperatures using the equations of Bemis et al. (1998) for AO, Grossman and Ku (1986) for GR | | | | | | | |
|---|---|---|---|---|---|---|---|---|---|---|---|---|---|---|---|---|---|
|  |  | Water $\delta^{18}O$ = 0.0 ‰ | | | | Water $\delta^{18}O$ = +0.4 ‰ | | | | Water $\delta^{18}O$ = 0.0 ‰ | | | | Water $\delta^{18}O$ = +0.4 ‰ | | | |
|  |  | SSFT | WSFT | SFR | SSR | SSFT | WSFT | SFR | SSR | SSFT | WSFT | SFR | SSR | SSFT | WSFT | SFR | SSR |
| M. | AO10 | 13.6 | 6.2 | 7.4 | 7.4 | 15.3 | 7.8 | 7.5 | 7.5 | 12.8 | 4.1 | 8.7 | 8.7 | 14.7 | 6.0 | 8.7 | 8.7 |
|  | AO9 | 15.9 | 6.2 | 9.7 | 9.7 | 17.6 | 7.8 | 9.9 | 9.9 | 15.4 | 4.1 | 11.3 | 11.3 | 17.3 | 6.0 | 11.3 | 11.3 |
| Oorderen | AO8 | 16.6 |  |  |  | 18.4 |  |  |  | 16.2 |  |  |  | 18.1 |  |  |  |
|  | AO7 | 18.0 | 5.9 | 12.1 | 15.1 | 19.8 | 7.4 | 12.4 | 15.4 | 17.7 | 3.6 | 14.1 | 17.1 | 19.6 | 5.6 | 14.1 | 17.1 |
|  | AO6 | 18.3 | 6.8 | 11.5 | 14.5 | 20.1 | 8.4 | 11.7 | 14.7 | 18.0 | 4.8 | 13.2 | 16.2 | 19.9 | 6.7 | 13.2 | 16.2 |
|  | AO5 | 18.6 | 5.8 | 12.8 | 15.8 | 20.4 | 7.3 | 13.1 | 16.1 | 18.3 | 3.5 | 14.8 | 17.8 | 20.3 | 5.4 | 14.8 | 17.8 |
|  | PR | 18.4 | 7.6 | 10.8 | 13.8 | 20.1 | 9.4 | 10.8 | 13.8 |  |  |  |  |  |  |  |  |
|  | AI | 18.2 | 6.0 | 12.2 | 15.2 | 19.9 | 7.7 | 12.2 | 15.2 |  |  |  |  |  |  |  |  |
| Luchtbal | AO4 | 16.2 | 6.6 | 9.6 | 12.6 | 18.0 | 8.2 | 9.8 | 12.8 | 15.8 | 4.5 | 11.2 | 14.2 | 17.7 | 6.5 | 11.2 | 14.2 |
|  | AO3 | 16.0 | 6.4 | 9.6 | 12.6 | 17.8 | 8.0 | 9.8 | 12.8 | 15.5 | 4.3 | 11.2 | 14.2 | 17.5 | 6.3 | 11.2 | 14.2 |
|  | AO2 | 15.9 | 7.0 | 8.9 | 11.9 | 17.6 | 8.6 | 9.1 | 12.1 | 15.4 | 5.0 | 10.4 | 13.4 | 17.3 | 6.9 | 10.4 | 13.4 |
|  | AO1 | 16.3 | 5.4 | 10.9 | 13.9 | 18.0 | 7.0 | 11.1 | 14.1 | 15.8 | 3.1 | 12.8 | 15.8 | 17.8 | 5.0 | 12.8 | 15.8 |
|  | GR2 | 16.4 | 11.1 | 5.4 | 8.4 | 17.5 | 12.1 | 5.4 | 8.4 | 17.9 | 9.1 | 8.8 | 11.8 | 19.6 | 10.8 | 8.8 | 11.8 |
|  | GR1 | 16.7 | 11.5 | 5.2 | 8.2 | 17.8 | 12.5 | 5.2 | 8.2 | 18.3 | 9.8 | 8.6 | 11.6 | 20.0 | 11.5 | 8.6 | 11.6 |




In conclusion, the data in Fig. 8a are probably the most accurate but the data in Fig. 8c should not be excluded from consideration, especially as evidence from modern *A. opercularis* (Johnson et al., 2021b) suggests that the equation of Bemis et al. (1998), used for calculation of temperatures from this species in Fig. 8c, provides more accurate temperatures than the equation of O'Neil et al. (1969), used in Fig. 8a.

### 6.1.2 Seasonal seafloor ranges

With the exception of the Luchtbal values resulting from the questionable combination of data employed in Fig. 8b, all the seasonal ranges for members (differences between mean summer and winter values for the various combinations of data in Fig. 8; Table 2) are lower than the current seafloor range at offshore locations in the southern North Sea—e.g. at 53° N, 03° E, where the range is 12.2 °C (Johnson et al., 2021b, fig. 1A). However, not all individual specimens indicate a lower range (Table 3): the *A. islandica* specimen (Oorderen Member) shows the same range as currently at 53° N, 03° E, and even a water $\delta^{18}O$ value of 0.0 ‰ used with the equation of O'Neil et al. (1969) yields a range (12.8 °C) from the Oorderen *A. opercularis* specimen AO5 that is higher than at present. The latter case is based on a winter temperature that is not reliable in the sense of Sect. 6.1.1, but the true (most extreme) winter temperature can only have been less and the seasonal range hence greater. Using a water $\delta^{18}O$ value of +0.4 ‰ with the equation of O'Neil et al. (1969) yields a range of 12.4 °C from the Oorderen specimen AO7, and using the equation of Bemis et al. (1998) yields a range of 13.2 °C from the Oorderen specimen AO6 and 12.8 °C from the Luchtbal specimen AO1 (independent of water $\delta^{18}O$), as well as higher ranges than with the equation of O'Neil et al. (1969) from the Oorderen specimens AO5 (14.8 °C) and AO7 (14.1 °C). It can therefore be said that at times during the period of deposition of the Oorderen Member the seasonal range in seafloor temperature was higher than at present, and possibly substantially so, and that the range may sometimes have been higher than at present during the period of deposition of the Luchtbal Member. Individual specimens from the equivalent of the Merksem Member provide no evidence of a higher seafloor range than now (maximum range 11.3 °C from AO9, calculated with the equation of Bemis et al., 1998), but the small sample size (two) should be noted.

The calculations leading to the above figures for seasonal range assume constant water $\delta^{18}O$ during the intervals of ontogeny concerned. If at the times of maximum temperature the actual water $\delta^{18}O$ value was lower than assumed the calculated temperatures would be overestimates;





similarly, if at the times of minimum temperature the actual water $\delta^{18}O$ value was higher than
assumed the calculated temperatures would be underestimates. Each of these situations, or the
two together, would yield an overestimate of seasonal range. While these circumstances are
possible, they are improbable. As noted in Sect. 2, water $\delta^{18}O$ is relatively high (not low) during
summer and relatively low (not high) during winter in coastal waters of the North Sea at
present. The calculated seasonal ranges are thus more likely to be underestimates.

**6.1.3 Seasonal surface ranges**

The water depths indicated by the bivalve mollusc assemblages of the Luchtbal and Oorderen
members (respective minimum depths 40 and 35 m; Sect. 3) are greater than the typical depth
of the summer thermocline in shelf settings (25–30 m). Microgrowth-increment evidence from
*A. opercularis* (Sect. 5.3) is consistent with a sub-thermocline setting for both members, and
the higher frequency of such evidence from the Luchtbal Member is consistent with the
indication from assemblage analysis that this was deposited at a greater depth than the
Oorderen Member. Microgrowth-increment evidence of a supra-thermocline setting from
Merksem-equivalent shells in the Oosterhout Formation at Ouwerkerk is similarly consistent
with the shallow depth of deposition (maximum 15 m) indicated by the biota of the Merksem
Member itself at Antwerp (note that supra-thermocline settings exist now in the southern North
Sea at a distance from the shore well beyond that of Ouwerkerk from the Pliocene shoreline;
Fig. 2). Given a sub-thermocline setting for the Luchtbal and Oorderen members we must add
a 'stratification factor' to summer seafloor temperatures to derive estimates of summer surface
temperature and hence seasonal surface range (winter surface temperature is likely to have been
the same as on the seafloor; Johnson et al., 2021b). There is a difference of 2.6 °C between the
annual seafloor and surface temperature maxima at a seasonally stratified location (depth 59
m) in the central North Sea some 600 km north of the sites of the Pliocene shells (Johnson et
al., 2021b). At this location the maximum surface temperature is only 13.7 °C, a figure
substantially exceeded in the (unstratified) southern North Sea at present (e.g. 17.1 °C at 53°
N, 03° E; Johnson et al., 2021b, fig. 1A) and with little doubt also in the Pliocene (see Sect. 3).
Therefore it can be scarcely contested that a stratification factor larger than 2.6 °C should be
employed. The northern Adriatic is probably a better modern analogue than the central North
Sea for the Pliocene SNSB. At a stratified location there (depth 38 m) annual seafloor and
surface temperature maxima differ by between 3.2 and 9.9 °C (mean difference 7.7 °C; Johnson
et al., 2021b), indicating that a stratification factor of 3 °C is the very least that should be
applied to derive estimates of seasonal surface range from Luchtbal and Oorderen seafloor





temperature data. Adding this figure to the seafloor ranges in Table 3 yields estimates of surface range that are larger than the present value in the southern North Sea (12.4 °C at 53° N, 03° E; Johnson et al., 2021b) from all individuals apart from the Luchtbal specimens GR1, GR2 and

AO2 (and a lower range is only obtained from the last when benthic temperature is calculated using the equation of O'Neil et al., 1969). Oorderen ranges are especially large. The low ranges from Luchtbal specimens GR1 and GR2 reflect the high winter seafloor temperatures recorded, all of which are unreliable, and AO2 would provide estimates of surface range no less than present with a stratification factor of 3.5 °C, a far from unreasonable figure. In contrast to the

evidence of a higher surface range than now at most times during the periods of deposition of the Luchtbal and Oorderen members, the two Merksem-equivalent shells show a lower seasonal range (Table 3), assuming they have been correctly interpreted as supra-thermocline individuals. It may of course be the case that they provide an unrepresentative picture of conditions during this interval.

**6.1.4 Absolute surface temperatures**

As pointed out in Sect. 6.1.1, there is a Luchtbal–Oorderen increase and an Oorderen–Merksem decrease in mean summer seafloor temperature from reliable individual summer values calculated using the equation of Royer et al. (2013) for *G. radiolyrata*, the equation of Grossman and Ku (1986) for *A. islandica* and *P. rustica*, and the equations of both O'Neil et

al. (1969) and Bemis et al. (1998) for *A. opercularis*. These changes are evident whether a water $\delta^{18}$O value of 0.0 ‰ or +0.4 ‰ is applied, are statistically significant, and correspond qualitatively to changes in summer temperature inferred from assemblages of ostracods and dinoflagellates. The data from ostracods and probably from dinoflagellates relate to summer surface temperature so quantitative comparison must involve the equivalent data, derived as

indicated in Sect. 6.1.3. With the exception of figures from AO8 derived using a water $\delta^{18}$O value of 0.0 ‰, all summer surface temperatures from Oorderen (and equivalent) individuals are above 20 °C (Table 4) and thus in agreement with dinoflagellate evidence specifying warm temperate conditions (summer temperature > 20 °C; Johnson et al., 2021b). Summer surface temperatures from Luchtbal (and equivalent) *A. opercularis* individuals are below 20 °C using

a water $\delta^{18}$O value of 0.0 ‰, but not +0.4 ‰, as are temperatures derived using the equation of Royer at al. (2013) from *G. radiolyrata* individuals (Table 4). Summer surface temperatures below 20 °C are in agreement with dinoflagellate evidence from the Luchtbal Member specifying cool temperate conditions (summer temperature < 20°C; Johnson et al., 2021b) and thus imply that the Oorderen (and equivalent) temperatures derived using a water $\delta^{18}$O value





of 0.0 ‰ are more accurate than the higher temperatures derived using a value of +0.4 ‰,

although it cannot be said which of the two equations applied to *A. opercularis* supplies the

most accurate figures. Using a water $\delta^{18}$O value of 0.0 ‰, the Oorderen (and equivalent) means

for summer surface temperature based on *A. opercularis* data are 21.0 ± 0.7 and 20.8 ± 0.8 °C

using the equations of O'Neil et al. (1969) and Bemis et al. (1998), respectively (Table 4). The

corresponding Merksem-equivalent means are 14.8 ± 1.2 and 14.1 ± 1.3 °C (cool temperate

values), signifying falls in summer surface temperature of 6.2 and 6.7°C, respectively. Very

similar or identical figures for this temperature decline (6.3 and 6.7 °C, respectively) are

obtained using a water $\delta^{18}$O value of +0.4 ‰. While one should not make too much of the

figures given the small Merksem-equivalent sample, the close similarity to the summer surface

temperature decline (5–6°C) inferred from ostracod assemblages should be noted.

**Table 4:** Summer (SSST) and winter (WSST) sea surface temperatures (°C; one d.p.) for the year showing the greatest seafloor range (cf. Table 3) from each shell, together with a figure (the midpoint between SSST and WSST values) for annual sea-surface temperature (ASST) from each shell and mean values for members and equivalents. SSST = summer seafloor temperature (SSFT; Table 3) for
Merksem-equivalent (Merk.) shells; SSST = SSFT + 3 °C for Luchtbal and Oorderen (and equivalent) shells (see Sect. 6.1.3 for explanation). ASST calculated from non-rounded SSST and WSST values and then rounded (one d.p.).

| Member and/or equivalent | Shell | Temperatures using the equations of O'Neil et al. (1969) for AO, Royer et al. (2013) for GR, Grossman and Ku (1986) for AI and PR | | | | | | Temperatures using the equations of Bemis et al. (1998) for AO, Grossman and Ku (1986) for GR, AI and PR | | | | | |
|---|---|---|---|---|---|---|---|---|---|---|---|---|---|
| | | Water $\delta^{18}$O = 0.0 ‰ | | | Water $\delta^{18}$O = +0.4 ‰ | | | Water $\delta^{18}$O = 0.0 ‰ | | | Water $\delta^{18}$O = +0.4 ‰ | | |
| | | SSST | WSST | ASST | SSST | WSST | ASST | SSST | WSST | ASST | SSST | WSST | ASST |
| Merk. | AO10 | 13.6 | 6.2 | 9.9 | 15.3 | 7.8 | 11.6 | 12.8 | 4.1 | 8.5 | 14.7 | 6.0 | 10.4 |
| | AO9 | 15.9 | 6.2 | 11.1 | 17.6 | 7.8 | 12.7 | 15.4 | 4.1 | 9.8 | 17.3 | 6.0 | 11.7 |
| | Mean | 14.8 ± 1.2 | 6.2 ± 0.0 | 10.5 ± 0.6 | 16.5 ± 1.2 | 7.8 ± 0.0 | 12.1 ± 0.6 | 14.1 ± 1.3 | 4.1 ± 0.0 | 9.1 ± 0.7 | 16.0 ± 1.3 | 6.0 ± 0.0 | 11.0 ± 0.7 |
| Oorderen | AO8 | 19.6 | | | 21.4 | | | 19.2 | | | 21.1 | | |
| | AO7 | 21.0 | 5.9 | 13.5 | 22.8 | 7.4 | 15.1 | 20.7 | 3.6 | 12.2 | 22.6 | 5.6 | 14.1 |
| | AO6 | 21.3 | 6.8 | 14.1 | 23.1 | 8.4 | 15.8 | 21.0 | 4.8 | 12.9 | 22.9 | 6.7 | 14.8 |
| | AO5 | 21.6 | 5.8 | 13.7 | 23.4 | 7.3 | 15.4 | 21.3 | 3.5 | 12.4 | 23.3 | 5.4 | 14.4 |
| | PR | 21.4 | 7.6 | 14.5 | 23.1 | 9.4 | 16.3 | 21.4 | 7.6 | 14.5 | 23.1 | 9.4 | 16.3 |
| | AI | 21.2 | 6.0 | 13.6 | 22.9 | 7.7 | 15.3 | 21.2 | 6.0 | 13.6 | 22.9 | 7.7 | 15.3 |
| | Mean | 21.0 ± 0.7 | 6.4 ± 0.7 | 13.9 ± 0.4 | 22.8 ± 0.6 | 8.0 ± 0.8 | 15.6 ± 0.4 | 20.8 ± 0.8 | 5.1 ± 1.5 | 13.1 ± 0.9 | 22.7 ± 0.7 | 7.0 ± 1.5 | 15.0 ± 0.8 |
| Luchtbal | AO4 | 19.2 | 6.6 | 12.9 | 21.0 | 8.2 | 14.6 | 18.8 | 4.5 | 11.7 | 20.7 | 6.5 | 13.6 |
| | AO3 | 19.0 | 6.4 | 12.7 | 20.8 | 8.0 | 14.4 | 18.5 | 4.3 | 11.4 | 20.5 | 6.3 | 13.4 |
| | AO2 | 18.9 | 7.0 | 13.0 | 20.6 | 8.6 | 14.6 | 18.4 | 5.0 | 11.7 | 20.3 | 6.9 | 13.6 |
| | AO1 | 19.3 | 5.4 | 12.4 | 21.0 | 7.0 | 14.0 | 18.8 | 3.1 | 11.0 | 20.8 | 5.0 | 12.9 |
| | GR2 | 19.4 | 11.1 | 15.3 | 20.5 | 12.1 | 16.3 | 20.9 | 9.1 | 15.0 | 22.6 | 10.8 | 16.7 |
| | GR1 | 19.7 | 11.5 | 15.6 | 20.8 | 12.5 | 16.7 | 21.3 | 9.8 | 15.6 | 23.0 | 11.5 | 17.3 |
| | Mean | 19.3 ± 0.3 | 8.0 ± 2.4 | 13.6 ± 1.3 | 20.8 ± 0.2 | 9.4 ± 2.1 | 15.1 ± 1.0 | 19.5 ± 1.2 | 6.0 ± 2.5 | 12.7 ± 1.8 | 21.3 ± 1.1 | 7.8 ± 2.4 | 14.6 ± 1.7 |

The similarity of $\delta^{18}$O-derived estimates of summer surface temperature, particularly those
using a water $\delta^{18}$O value of 0.0 ‰, to assemblage-based estimates lends credence to the

equivalent winter surface temperatures, which are nearly all firmly in the cool temperate range

(see discussion of discrepant data from *G. radiolyrata* below). We can take the midpoint

between the summer and winter temperatures from individuals as a figure for annual (average)



sea-surface temperature and compare interval means with MASST estimates based on other

information. Robinson et al. (2018) estimated a MASST of 13.6 °C for the mid-Piacenzian North Sea on the basis of bivalve $\delta^{18}O$ data from the Coralline Crag Formation (UK). It, is, however, questionable whether the Coralline Crag is of this age (see Fig. 3). Dearing Crampton-Flood et al. (2020) obtained a figure of about 16 °C (cool temperate) for summer surface temperature from alkenone index and  a figure of about 10 °C (boundary cool/warm

temperate) for winter surface temperature from TEX$_{86}$ for part of the Oosterhout Formation in the Netherlands (Noord-Brabant) representing the MPWP. On the above basis these figures specify a MASST of 13°C, a temperature very similar to the $\delta^{18}O$-derived estimates (MASST values of 13.1 ± 0.9 and 13.9 ± 0.4 °C, using a water $\delta^{18}O$ value of 0.0 ‰; Table 4) from Oorderen and equivalent shells, which represent the same interval. Such figures for MASST

(2–3 °C higher than the figure of 10.9 °C obtained from modern summer and winter surface temperatures of 17.1 and 4.7 °C, respectively, at 53° N, 03° E; Johnson et al., 2021b, fig. 1A) are entirely consistent with general expectations for the MPWP but the $\delta^{18}O$-derived data reveal that they result from substantially warmer summer conditions than at present and winter conditions much the same as now, opposite to the picture provided by alkenone and TEX$_{86}$

data. The $\delta^{18}O$-derived data from Luchtbal and equivalent shells similarly indicate warmer summer conditions than at present, although less markedly so. The Luchtbal figures for mean winter surface temperature (and MASST) in Table 4 are undoubtedly overestimates because they incorporate truncated winter data from *G. radiolyrata*: the figures based on *A. opercularis* shells alone (6.4 ± 0.6 °C and 4.2 ± 0.7 °C using a water value of  0.0‰) are very similar to

those from all Oorderen and equivalent shells. The Merksem-equivalent shells yield similarly low winter temperatures but summer temperatures well below those indicated by Oorderen and equivalent shells, and below those typical of the southern North Sea at present, though it has to be acknowledged that the two shells investigated may not be representative of their time.

## 6.2 Meaning of δ¹³C data

The ontogenetic decline in $\delta^{13}C$ shown by *A. opercularis* is as seen in modern examples of the species from diverse settings (Johnson et al., 2021b), and probably reflects increasing output of isotopically light respiratory carbon with increasing body size alongside slower shell secretion—i.e. reduced 'demand' for carbon (Lorrain et al., 2004). Short-term fluctuations paralleling changes in $\delta^{18}O$ might similarly reflect variation in respiratory output determined

by seasonal variation in the availability of food (Chauvaud et al., 2011). The opposite ontogenetic trend shown by *G. radiolyrata* and *A. islandica* can hardly be explained by a



reduction in respiratory output, and the opposite short-term pattern shown by *G. radiolyrata* is unlikely to reflect a reversal in the timing of maximum food availability from summer to winter. These changes might reflect variation in the $\delta^{13}C$ of dissolved inorganic carbon (DIC), the main

source of carbon in shells (Marchais et al., 2015). Preferential uptake of $^{12}C$ by photosynthesizers is a major influence on the $\delta^{13}C$ of DIC, but high photosynthetic fixation of carbon in summer is a doubtful cause of the high summer $\delta^{13}C$ values in *G. radiolyrata* because it would require (cf. Arthur et al., 1983) that the shells lived above the thermocline, which other evidence argues against. The anomalously low $\delta^{13}C$ values from *P. rustica* (for the aragonite

mineralogy) cannot be explained as a consequence of the incorporation of isotopically light carbon from porewaters (cf. Krantz et al., 1987) because the other infaunal species, *G. radiolyrata* and *A. islandica*, exhibit high values. Conceivably, the *P. rustica* values reflect a food source with a particularly low value (Marchais et al., 2015).

In view of the multiple potential 'local' controls on shell $\delta^{13}C$ it is questionable whether the relatively low values from late Pliocene *A. opercularis*, compared to pre-industrial Holocene examples (Hickson et al., 2000), are a reflection of relatively high atmospheric $CO_2$, as was suggested to explain the similarly low values from early Pliocene forms (Johnson et al., 2009; Vignols et al., 2019).

**7 Implications of temperature data**

We have shown that by adopting certain equations relating shell $\delta^{18}O$ to temperature, selecting a particular modelled value for water $\delta^{18}O$, and making a reasonable allowance for summer stratification (where indicated by independent evidence), it is possible to generate summer surface temperatures from shell $\delta^{18}O$ data that are consistent with assemblage-derived

estimates (cool or warm temperate according to interval) for the late Pliocene of Belgium and the Netherlands. The corresponding winter surface temperatures are cool temperate in each of the three (Luchtbal, Oorderen and Merksem) intervals studied, and in conjunction with the summer temperatures demonstrate a higher seasonal surface range than at present in the area during the Luchtbal and Oorderen intervals, and a particularly high range during the latter. The

same approach has also revealed high seasonal surface ranges in the early Pliocene of the SNSB and the early and late Pliocene of the eastern seaboard of the USA (Johnson et al., 2017, 2019, 2021b; Vignols et al. 2019). Southward-flowing cool currents, as exist now (north of Cape Hatteras), were probably influential in the latter area, but no such current exists at present in





the North Sea or is likely to have done during the Pliocene. Presently, winter temperature is
raised somewhat in the North Sea by offshoots of the warm North Atlantic Current, principally
entering from the north (Winther and Johannessen, 2006). Reduction of this oceanic heat
supply, in conjunction with global (atmospheric) warmth, might perhaps have led to the
seasonal surface temperatures of the Luchtbal and Oorderen intervals (similar to now in winter,
warmer than now in summer) that account for the enhanced seasonal ranges. Fluctuations in
the strength and position of the North Atlantic Current during the Pliocene are certainly
recognized from proxy evidence, and episodes of reduced oceanic heat supply could
correspond to the Luchtbal and Oorderen intervals (e.g. Bachem et al., 2017; Panitz et al.,
2018). For the latter (i.e. the MPWP), when seasonality was greatest in the SNSB, most models
indicate an increase in ocean heat transport in the North Atlantic compared to now, but some
indicate a decrease (Zhang et al., 2021). However, even if the times of high seasonality in the
SNSB correspond to periods of reduced oceanic heat supply (relative to the Pliocene norm or
to the present) it is not yet clear that this is a sufficient explanation for the low winter
temperatures contributing to pronounced seasonality. The Merksem decline in summer
temperature (post-dating the MPWP) can be more certainly attributed to the global cooling
which presaged the onset of northern hemisphere glaciation. The lack of a decline in winter
temperature is consistent with Mg/Ca evidence from North Atlantic *Globigerina bulloides*
(Foraminifera) suggesting that minimum temperatures did not start falling until the very end of
the Pliocene, after the time of deposition of the Lillo Formation (Hennissen et al., 2015, 2017).

The similarity of the alkenone/TEX$_{86}$-based estimate of MASST for the MPWP in the eastern
SNSB to the sclerochronologically derived estimates both corroborates the latter and suggests
that alkenone temperatures for other areas could be usefully supplemented by information from
TEX$_{86}$, even if this would be unlikely to specify the full seasonal range. Combined (average)
figures would probably be lower than alkenone-only estimates and the relatively close
alignment of proxy with model temperatures recently achieved for the northern North Atlantic
(Sect.1) might be lost, with implications for model adequacy. Alkenone temperatures can also
be checked against sclerochronologically derived estimates for the MPWP in the US Middle
Atlantic Coastal Plain. Here, as in the eastern SNSB, summer surface figures projected from
benthic isotopic temperatures (Johnson et al., 2017, 2019) are several degrees higher than
alkenone temperatures, which are typically 25–26 °C (Dowsett et al., 2019). However, again
like the eastern SNSB, winter isotopic temperatures are very much cooler (at least from scallop
species) and specify a MASST in the region of 20 °C. Thus sclerochronological evidence from





the Middle Atlantic Coastal Plain also suggests that alkenone temperatures should be 'moderated' by combination with winter proxy data in order to facilitate meaningful comparison with model results.

The summer seafloor temperatures recorded from a late Pliocene *A. islandica* specimen herein (18.2 and 19.9 °C using water $\delta^{18}O$ values of 0.0 and +0.4 ‰, respectively) corroborate evidence supplied by an early Pleistocene example from Italy (20.3 °C from specimen ACG254-1 using a water $\delta^{18}O$ value of +0.5 ‰; Crippa et al., 2016, table 1) of a thermal range greater in the past than at present (upper limit 16 °C; Witbaard and Bergman, 2003). This indication of change in realized thermal niche supplements evidence of the same in several other bivalve taxa from the early Pliocene of the UK, although in most of these cases it is manifested by tolerance then of winter conditions cooler than are experienced by modern representatives or relatives, which are restricted to Mediterranean locations (Vignols et al., 2019). This information raises some doubts about the use of assemblage evidence to interpret past environments by means of ecological uniformitarianism, the very widely applied approach which assumes that ancient examples of taxa occupied the same niche as modern. Certainly the accuracy of this methodology for times beyond the recent past deserves reconsideration.

**8 Further work**

Even if the analysed shells of Luchtbal and Oorderen age were to be from supra- rather than sub-thermocline settings, the $\delta^{18}O$ data from them would specify a seasonal surface temperature range much higher than previously inferred—for example, most shells of Oorderen age indicate a range more than double the 6 °C suggested by organic proxies (see Sect. 6.1.4 and the SFR data in Table 3). We argued in Sect. 2 and Sect. 6.1.2 that seasonal variation in shell $\delta^{18}O$ is unlikely to have been enhanced by variation in water $\delta^{18}O$, and it can be added here that fluvial input (the means by which variation in water $\delta^{18}O$ might have been brought about) may have been less in the Pliocene than now due to a smaller catchment area of the Rhine/Meuse/Scheldt system (Dearing Crampton-Flood et al., 2020), making enhancement of shell $\delta^{18}O$ variation by water $\delta^{18}O$ variation even more improbable. Notwithstanding these arguments, actual evidence of water $\delta^{18}O$ would be very welcome, both as a check on the stability of values through the year and as a means of deriving accurate absolute temperatures. At present, substitution of independently derived temperatures (e.g. from carbonate clumped isotope or biomineral unit thermometry; Briard et al., 2020, Höche et al. 2020, 2021;





Caldarescu et al., 2021; de Winter et al., 2021) into equations relating shell $\delta^{18}O$ to temperature
and water $\delta^{18}O$ seems the best approach to determining the last parameter. However, the
existence of fluid inclusions in bivalve shell carbonate (Nooitgedacht et al., 2021) raises the
possibility that these might serve as a source of direct insight into the isotopic composition of
ambient water during life.


It would, of course, also be useful to have additional shell $\delta^{18}O$ (and increment) data, more to
confirm the decline in summer sea-surface temperature, annual sea-surface temperature and
sea-surface range indicated by the limited information from Merksem-age shells than to expand
the already substantial evidence of high values for these parameters from shells of Luchtbal

and Oorderen age. Notwithstanding some doubts about the reliability of the approach (Johnson
et al., 2017), it would also be worth obtaining independent evidence of seasonality from
variation in the size of zooids within bryozoan colonies. Using this technique, Knowles et al.
(2009, table 3) obtained locality-specific seasonality estimates of 8.08 ± 1.38 and 8.15 ± 1.30
°C for the early Pliocene Ramsholt Member of eastern England, figures comparing closely with

the value of 7.77 ± 1.12 °C obtained through isotope sclerochronology of *A. opercularis* from
this unit (mean difference between the winter and summer seafloor temperatures in Johnson et
al., 2021b, table 3).

In addition to the above refinements of proxy evidence, further modelling efforts are required

to see whether the low winter temperatures indicated by sclerochronological evidence for the
mid-Piacenzian SNSB can be reproduced, and whether reduced oceanic heat supply is a
feasible cause. Haywood et al. (2000) modelled mid-Piacenzian summer surface temperatures
2–4 °C higher than present for the area, similar to those determined herein. However, they
modelled winter temperatures 4–6 °C higher than present, approaching or into the warm

temperate range and markedly above the firmly cool temperate values indicated by
sclerochronology. Haywood et al. (2000) ascribed the reduced seasonality specified by their
results to greater westerly wind stress and strength in the North Atlantic compared to now, with
a resultant increase in heat transport by the Gulf Stream and North Atlantic Current. More
recent modelling of mid-Piacenzian seasonal sea-surface temperatures at higher northern

latitudes indicates greater warming in summer than winter (de Nooijer et al., 2020)—i.e. higher
seasonality than now, as inferred for the SNSB. Moreover, as already noted (Sect. 7), oceanic
heat supply may not have been greater in the mid-Piacenzian. Possibly, use of up-to-date



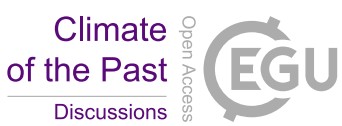

models with revised boundary conditions may yield results conforming with the evidence of low winter temperatures and high seasonality from the SNSB.

**9 Conclusions**

Sclerochronological evidence from bivalves indicates that for most of the late Pliocene (including the MPWP) the seasonal range in surface temperature in the SNSB was higher than now. This was probably a consequence of higher summer temperatures associated with global (atmospheric) warmth. The apparently similar winter temperatures to now may reflect partial

withdrawal of oceanic heat supply to the region through a change in strength and/or position of the North Atlantic Current.

Averaging sclerochronologically derived summer and winter temperatures yields a MASST 2–3°C higher than now in the SNSB, as does averaging temperatures from alkenone and TEX$_{86}$

thermometry. However, alkenones provide underestimates of extreme summer temperature and TEX$_{86}$ provides overestimates of extreme winter temperature, hence these proxies do not specify the full seasonal range. The sclerochronologically derived temperatures are based on shell $\delta^{18}$O and dependent on estimates of water $\delta^{18}$O that require testing. Back-calculation from temperatures obtained by carbonate clumped isotope or biomineral unit thermometry from the

same shells constitutes a potential means.

**Data availability.** The raw isotope and increment data, and the corresponding shell heights, are  available  in  the  open-source  online  repository  Zenodo (https://doi.org/10.5281/zenodo.5585630; Johnson et al. 2021c).

**Author contributions.** ALAJ conceived the study, obtained financial support and access to

isotope analytical facilities, conducted some of the isotopic sampling and treatment of results, and drafted the manuscript. AMV conducted the rest of the isotopic sampling and treatment of results, and obtained the increment data, within the context of a PhD project supervised by ALAJ. BRS and MJL facilitated the isotopic analysis. SG provided shell photographs together with detailed information concerning the stratigraphy and environments of the Belgian and

Dutch Pliocene, supplied as comments on the first draft of the manuscript. BRS and MJL also commented on the first draft.



**Competing interests.** The authors declare that they have no conflict of interest.

**Acknowledgements.** We thank the following for facilitating study of museum specimens in their care: Annelise Folie, Robert Marquet and Etienne Steurbaut (Institut royal des Sciences
naturelles de Belgique, Brussels, Belgium); Frank Wesselingh (Naturalis Biodiversity Center, Leiden, The Netherlands); Serge Gofas and Virginie Héros (Muséum National d'Histoire Naturelle, Paris, France). Melita Peharda (Institut za oceanografiju i ribarstvo, Split, Croatia) and Guillermo Roman (Instituto Español de Oceanografía, La Coruña, Spain) generously supplied live-collected individuals. Mark Dean and Matthew Hunt (Geoscience, University of
Derby, UK) kindly assisted with specimen preparation, and Michael Maus (Institut für Geowissenschaften, Universität Mainz, Germany) and Hilary Sloane (National Environmental Isotope Facility, British Geological Survey, Keyworth, UK) with isotopic analysis. We are grateful to the NERC Isotope Geosciences Facilities Steering Committee for granting analytical services (IP-1108-0509, IP-1155-1109), the British Geological Survey for a PhD studentship
award to AMV (BUFI S157), and the Alexander von Humboldt-Stiftung for support of a research stay at Mainz by ALAJ.

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
