# Peer review of "Sclerochronological evidence of pronounced seasonality from the late Pliocene of the southern North Sea Basin, and its implications"

_Climate of the Past, 2021_

## Author Comment (AC1)

[Figure]

**Figure 8:** Winter and summer seafloor temperatures, calculated from the seasonal extreme $\delta^{18}O$ values indicated in Figs. 6 and 7, using water $\delta^{18}O$ values of 0.0 ‰ and +0.4 ‰ and various equations. **(a)**: equation of Royer et al. (2013) for GR, of O'Neil et al. (1969) for AO, and of Grossman and Ku (1986) for AI and PR; **(b)**: equation of Grossman and Ku (1986) for GR, AI and PR, and of Bemis et al. (1998) for AO; **(c)**: equation of Royer et al. (2013) for GR, of Bemis et al. (1998) for AO, and of Grossman and Ku (1986) for AI and PR. Interval means are for reliable seasonal temperatures (see Sect. 6.1.1) from the Luchtbal Member and equivalent, Oorderen Member and equivalent, and Merksem-equivalent strata. The indicated present-day seasonal seafloor temperature range (4.7–16.9 °C) is for 25 m depth at 53° N, 03° E. Note that Pliocene ranges, as indicated by the separation of dotted or dashed lines for each interval, are broadly similar to the present-day range (see text for more detailed discussion).

[Figure]

**Figure 9:** Winter and summer sea-surface temperatures, calculated as in Fig. 8, using the same equations for **(a)**, **(b)** and **(c)**, but with a 3 °C supplement to Luchtbal and Oorderen summer temperatures in recognition of thermal stratification (see text for explanation). Interval means calculated as in Fig. 8. The indicated present-day seasonal sea-surface temperature range (4.7–17.1 °C) is for 53° N, 03° E. Note that the Luchtbal and Oorderen ranges, as indicated by the separation of dotted or dashed lines for each interval, are larger than the present-day range (see text for more detailed discussion).

---

## Author Response (AR1)

**I have pasted in below all the comments from the editor, the reviewers and Niels de Winter (CC), together with my initial responses (black bold type). I have essentially done everything indicated in the initial responses, and also made some minor rewordings, corrections and additions (including references). The final responses (red bold type) indicate how and where the initial responses have been implemented. Please note that Figs 4 and 5 are entirely original and hence do not require crediting to any source (I answered a query about this from the Copernicus team some while ago but it remains on the system).**

Editor's comments

Based on the reviewers' comments and your responses I would like to invite you to submit a revised version of the manuscript, where your responses to the reviewers' comments are implemented. Please make it visible through track-changes where and what changes you have done in the revised manuscript.

**Done in the 'tracked changes' document supplied.**

I agree with Reviewer 1 that Fig. 6 and 7 will be easier to read if you increase the size of the font at the axes/axes descriptions. This can be done without changing the size of the full figure and it will make much easier to read. Similarly, I agree (as you also do in your response) that the readability of Table 3 will benefit from being presented in landscape mode.

**We have increased the size of the units and axis titles in Figs 6 and 7 by four points. The content, format and number of tables (there are now five) has changed a little (see response to CC1). The larger ones (Tables 3 and 5) may well benefit from being produced 'side on' (landscape format) in the published version, but they have been included in portrait format in the integrated file for review.**

(RC1 comments in normal typeface; **responses in bold**)

The paper "Sclerochronological evidence of pronounced seasonality from the late Pliocene of the southern North Sea Basin, and its implications» by Andrew L. A. Johnson et al., aims at reconstructing the late Pliocene seasonal range of the seafloor and sea surface temperature in the southern North Sea Basin.

This topic is relevant for CP. The manuscript is clear and well-written, it presents new data, and substantial conclusions are reached.

The data used in the paper are primarily based on stable oxygen isotope analysis of growth increments in different bivalve species from formations in Belgium and in the Netherlands.

When using sub-fossil shells for climate reconstructions, there are several issues that need to be addressed when drawing conclusions about the seasonality of water

temperature. These are the overall differences in d18O seawater between the Pliocene and today, possible seasonal variations in water d18O, uncertainties depth habitat (above/below the thermocline), aliasing of the d18O signal in relationship to ontogenetic decreases in growth, estimates of the changes in temperature between surface and bottom during the late Pliocene, and possible differences in the thermal niche between the late Pliocene and today.

The authors do a good job addressing the possible implications of these uncertainties on the reconstructed seasonal ranges in temperature. For the present paper, it seems likely that the biggest unknowns are the depth habitat and, for the calculation of absolute sea-surface temperatures, the estimate of summer stratification. However, the authors go through and reason around these uncertainties in detail. The authors also discussed briefly possible shifts in the thermal niche over time, which is an important point.

The interpretation of the 13C signal in the shells is not entirely necessary for the main story. However, given the complex interpretation of other isotopic data and sclerochronological analyses in the paper, I think that many readers will appreciate that the d13C data are also addressed, at least to some extent.

**The referee is quite right that interpretation of the $\delta^{13}$C data (obtained alongside $\delta^{18}$O) is not essential to the main (temperature) story. In so far as atmospheric CO$_2$ influences global temperature we thought it worthwhile to make the point that the relatively low $\delta^{13}$C values from Pliocene *Aequipecten opercularis* are rather doubtfully a reflection of the high atmospheric CO$_2$ (independently indicated) at that time, as had been suggested previously. In order to make this point we needed to show the more likely (local) controls on shell $\delta^{13}$C. We did so as concisely as possible to avoid a lengthy departure from the main story.**

**No further response required.**

Some graphs (6 and 7) are tiny, and not entirely suitable for the aging eye. The same goes for the tables. Is it possible to present the comparison of temperature ranges (Pliocene vs modern) in a graph?

**Figs. 6 and 7 were designed with a view to each occupying the full width of a page, like similar diagrams in Johnson et al. (2009, 2017, 2019, 2021). If reproduced in this way they will be considerably larger than they appear in the pre-print. It will also be possible to reproduce the tables at a larger size than in the pre-print, though Tables 1 and 3 might be best 'side on' (in landscape format), on their own pages. With the addition prompted by RC2, Fig. 8 now enables comparison of Pliocene with modern seafloor temperature ranges, and a new figure (Fig. 9) has been created in the same format to enable comparison of surface temperature ranges. Both figures are attached for perusal.**

**See response to editor and the revised integrated file, containing Figs 8 and 9.**

All in all, this is a nice paper that is an interesting read to many, especially sclerochronologists working on past warm climates.

**References**

Johnson, A.L.A., Hickson, J.A., Bird, A., Schöne, B.R., Balson, P.S., Heaton, T.H.E. and Williams, M., 2009. Comparative sclerochronology of modern and mid-Pliocene (c. 3.5 Ma) *Aequipecten opercularis* (Mollusca, Bivalvia): an insight into past and future climate change in the north-east Atlantic region. Palaeogeography, Palaeoclimatology, Palaeoecology 284, 164–179. https://doi.org/10.1016/j.palaeo.2009.0.022.

Johnson, A.L.A., Valentine, A., Leng, M.J., Sloane, H.J., Schöne, B.R. and Balson, P.S., 2017. Isotopic temperatures from the Early and Mid-Pliocene of the US Middle Atlantic Coastal Plain, and their implications for the cause of regional marine climate change. Palaios 32, 250–269. https://doi.org/10.2110/palo.2016.080.

Johnson, A.L.A., Valentine, A.M., Leng, M.J., Schöne, B.R. and Sloane, H.J., 2019. Life history, environment and extinction of the scallop *Carolinapecten eboreus* (Conrad) in the Plio-Pleistocene of the US eastern seaboard. Palaios 34, 49–70. https://doi.org/10.2110/palo.2018.056.

Johnson, A.L.A., Valentine, A.M., Schöne, B.R., Leng, M.J., Sloane, H.J. and Janeković, I., 2021. Growth-increment characteristics and isotopic ($\delta^{18}O$) temperature record of sub-thermocline *Aequipecten opercularis* (Mollusca:Bivalvia): evidence from modern Adriatic forms and an application to early Pliocene examples from eastern England. Palaeogeography, Palaeoclimatology, Palaeoecology 561, article 110046. https://doi.org/10.1016/j.palaeo.2020.110046.

(RC2 comments in normal typefaces; **responses in bold**)

General Comments

Johnson and co-authors discuss seasonality, an under-investigated but essential dynamic in palaeoclimatology, from the southern part of the North Sea Basin (SNSB) during the last episode in Earth History when global climate was consistently warmer than today. They use stable isotope measurements on benthic marine molluscs (*Aequipecten opercularis*, *Pygocardia rustica*, *Arctica islandia* and *Glycymeris radiolyrata*) sourced from the Luchtbal, Oorderen and Merksem Members (and their lateral equivalents) from the Lillo Formation in North Belgium and the Southern Netherlands. The authors use the extreme inflection points of the δ18O ontogenetic profiles from the recovered molluscs to compute summer and winter seafloor temperatures using various equations against an assumed background of 0.0–0.4‰ average δ18Osw in the SNSB. The derived temperature difference taken at these inflection points is interpreted as a seasonality signal. Johnson et al. conclude that seasonality of the late Pliocene SNSB was on average more pronounced than it is now (3°C higher). Summer temperatures were found to be higher in the late Pliocene while winter temperatures were comparable to today.

I believe the subject matter is relevant to the diverse readership of Climate of the Past and closely matches the scientific remit of the journal. The manuscript is well-structured, underpinned by clear, objective and very precise writing. As a result, it is easy to follow the employed methodology, the authors' interpretation of the results and how the conclusions were reached. The text is supported by a set of informative figures which

are, despite the complexity of the incorporated data, kept simple and straightforward to interpret.

The authors present a very honest assessment of their results, easily exceeding what can be reasonably expected as the baseline for scientific scrutiny. This is exemplified by Figure 8 where the authors compare different computation methods in the literature and select the most suitable algorithm for their specific case.

Specific comments

*The age of the different members of the Lillo Formation was constrained using biostratigraphy and sequence stratigraphy: Luchtbal Member (3.71–3.21 Ma), Oorderen Member (3.21–2.76 Ma) and the Merksem Member (3.21–2.76) (Figure 3 and De Schepper et al., 2009). This covers the mid-Piacenzian Warm Period, but also includes glacial events like MIS M2 (De Schepper et al., 2013), with a total fluctuation of 0.89‰ in the orbitally-tuned global stack of benthic δ18O (Lisiecki and Raymo, 2005) over this period. Evidence suggests that seasonality in proxy records is more pronounced in colder, glacial conditions (Crippa et al., 2016; Hennissen et al., 2015 and references therein). Is there a way of tying the seasonality results from the current study into the global climatic picture or should they be viewed as endemic snapshots of seasonality of the late Pliocene (which may be against a background which could range from MIS M2 to MPWP)? Are the reported seasonality values to be viewed as an average seasonality signal for the late Pliocene or is it more of a minimum estimate?

**The age evidence as supplied by De Schepper et al. (2009) is indeed as stated but we incorporated the additional evidence of Louwye and De Schepper (2010), which indicates that the upper boundary of the Oorderen Member is no younger than 3.15 Ma – i.e. that this unit was entirely deposited within the Mid-Piacenzian Warm Period (MPWP). We will correct our omission of Louwye and De Schepper (2010) from the list of sources given in the caption to Fig. 3. It is of course true that there were fluctuations in deep-sea benthic δ18O during the MPWP, and that these (in the order of 0.3 ‰ either side of the modern value) signify relatively cool and warm intervals. It seems reasonable to assume that the few horizons in the Oorderen Member with a 'cool' dinoflagellate biota represent the former and that the more numerous horizons with a 'warm' biota, including the horizon (*Atrna fragilis* bed) supplying five of the six Oorderen-Member specimens used in this study, represent the latter. From the evidence that seasonality is higher under cooler conditions these specimens can be viewed as providing a minimum estimate of average MPWP seasonality. The 3.71–2.76 Ma age limits for the Luchtbal Member encompass the MIS M2 glacial at c. 3 Ma. In so far as the projected mean sea-surface temperature from this unit is lower than from the Oorderen Member it may be that it was deposited under glacial conditions. However, the difference is only small and seasonality was no greater so it seems more likely that the data from the Luchtbal Member are representative of some part of the long, predominantly warm interval extending back from MIS M2 to 3.71 Ma. De Schepper et al. (2009) interpreted the unconformity above the Luchtbal Member as a consequence of the sea-level lowering associated with the MIS M2 glacial, confirming the view that the unit represents some part of the interval indicated. We will include at least some of this more refined discussion of the data in the revised version of the manuscript.**

**The reference to Louwye and De Schepper (2010) has been included, the age and setting (glacial/interglacial) of the Luchtbal Member has been discussed (LL360–363), and the implications for interpretation of seasonality have been explained (LL908–911). The more general issue of comparability of the data to information from other sources is discussed in the next response.**

\*Sclerochronologically derived temperature estimates offer an invaluable window into the (sub)annual temperature fluctuations that the biotic carriers were exposed to. Other techniques (e.g. foraminiferal Mg/Ca, alkenones and TEX86) offer estimates that are averaged over much longer time periods. Do these differences in temporal resolution complicate cross-proxy comparison? Can the sclerochronologically derived results be viewed as a tool to set the true range of seasonality recorded in other proxies that cannot capture this accurately but have the advantage of stretching observations over larger intervals?

**Our isotope-based sclerochronological approach yields a very similar mean annual sea-surface temperature for the MPWP in the SNSB to that derived by averaging the alkenone and $TEX_{86}$ temperatures of Dearing Crampton-Flood et al. (2020). We therefore agree with these authors that alkenone and $TEX_{86}$ temperatures represent seasonal values (respectively, summer and winter) but conclude from comparisons with our seasonal sea-surface temperatures that they do not signify the seasonal extremes and hence yield an underestimate of the seasonal range. This difference in range estimate will persist when we compare alkenone/$TEX_{86}$ seasonality with our figure derived from mean (rather than individual) summer and winter sea-surface temperatures for the relevant stratigraphic unit (Oorderen Member), as intended in the revised version of the manuscript (see the response to CC1). The figure derived from mean values is formally comparable with the alkenone/$TEX_{86}$ figure in that it integrates data over an uncertain but undoubtedly lengthy interval. However, even if annual average temperature fluctuated during this interval (highly likely on the evidence of instrumental data for recent centuries) this integrated figure is likely to give a good insight into seasonality in individual years, as indeed the present data show (seasonality from the pooled data is similar to that from each shell). As well as providing a more accurate picture of seasonality, the sclerochronological approach, applied to long-lived shells, can in principle provide a record of year-by-year variation in annual average temperature which is not recoverable from time-averaged data (as from alkenones, $TEX_{86}$ and Mg/Ca of multiple forams). Unfortunately, as the records from _Glycymeris radiolyrata_ in this study show, long-lived shells sometimes do not give a full picture of seasonal temperatures and in such cases cannot supply an accurate figure for annual average temperature. We will again include at least some of this discussion in the revised version of the manuscript. However, we do not want to stray too far into interpreting results from other proxies for fear that it might reduce the coherence of the interpretation of sclerochronological data.**

**As pointed out above, our focus on mean temperatures in the revised version addresses the issue of comparability. We have made specific mention of this at LL819–821. The sections of text where we now focus on mean temperatures are identified under the response to the comments of Niels de Winter (CC).**

Technical corrections

Line 460: measurements were made in two different laboratories and analytical errors were reported. Were replicate samples run to assess the inter-laboratory variation?

**No individual shell samples were analyzed at both Keyworth and Mainz but Johnson et al. (2019) recorded the following in relation to analytical results from the two laboratories: 'For a few shells, part of the sample series was analyzed in one laboratory and part in the other; there was found to be excellent agreement (e.g., smooth continuation of trends) between the subsets of data.' This could be referenced as evidence of the comparability of results from each lab, but the existing statement about results from analysis of NBS-19 covers this point (lines 471–472). For the present study, all samples from each shell were analysed either in one laboratory or the other (see lines 460–462).**

**We have referred (LL485–486) to Johnson et al. (2019).**

Line 532: insert comma after 'cycle'.

**OK.**

**Done (L550).**

Line 573: is there a way of quantifying the covariation?

**We will supply $R^2$ values (see also the response to CC1).**

**Done (LL592, 594).**

Figure 8: it may be informative to put in a vertical line (grey in background maybe) to indicate current summer and winter temperatures as a direct comparison to the measurements and calculations in each panel.

**Rather than add further vertical lines to those already present we have chosen to depict the modern seasonal range by a grey bar of appropriate width, the seafloor range being shown in Fig. 8 and the fractionally larger surface range in Fig. 9 (new figure; see the response to RC1). Both figures are attached for perusal.**

**Done (Figs 8, 9).**

Table 3: the text is rather small and it may be better to put the entire table in landscape format.

**Agreed. As pointed out in the response to RC1, it will be possible to reproduce the table (and the text within it) at a larger size than in the pre-print, without altering its orientation. However, a 'side-on' (landscape) format may be better.**

**See response to editor.**

In conclusion, I believe this paper is an example of how ecological uniformitarianism can be employed to evaluate palaeoclimatological conditions and offer invaluable constraints for climate models.

References

Crippa, G., Angiolini, L., Bottini, C., Erba, E., Felletti, F., Frigerio, C., Hennissen, J.A.I., Leng, M.J., Petrizzo, M.R., Raffi, I., Raineri, G., Stephenson, M.H., 2016. Seasonality fluctuations recorded in fossil bivalves during the early Pleistocene: Implications for climate change. Palaeogeog. Palaeoclimatol. Palaeoecol. 446, 234-251.

De Schepper, S., Groeneveld, J., Naafs, B.D.A., Van Renterghem, C., Hennissen, J., Head, M.J., Louwye, S., Fabian, K., 2013. Northern Hemisphere Glaciation during the Globally Warm Early Late Pliocene. PLoS ONE 8, e81508.

De Schepper, S., Head, M.J., Louwye, S., 2009. Pliocene dinoflagellate cyst stratigraphy, palaeoecology and sequence stratigraphy of the Tunnel-Canal Dock, Belgium. Geological Magazine 146, 92.

**Additional references**

**Dearing Crampton-Flood, E., Noorbergen, L.J., Smits, D., Boschman, R.C., Donders, T.H. and Muns, D.K., 2020. A new age model for the Pliocene of the southern North Sea basin: amulti-proxy climate reconstruction. Climate of the Past 16, 523–541. https://doi.org/10.5194/cp-16-523-2020.**

**Johnson, A.L.A., Valentine, A.M., Leng, M.J., Schöne, B.R. and Sloane, H.J., 2019. Life history, environment and extinction of the scallop Carolinapecten eboreus (Conrad) in the Plio-Pleistocene of the US eastern seaboard. Palaios 34, 49–70. https://doi.org/10.2110/palo.2018.056.**

**Louwye, S. and De Schepper, S., 2010. The Miocene–Pliocene hiatus in the southern North Sea Basin (northern Belgium) revealed by dinoflagellate cysts. Geological Magazine 147, 760–776. https://doi.org/10.1017/S0016756810000191.**

(CC1 comments in normal typeface; **responses in bold**)

In their manuscript, Johnson and colleagues present a nice dataset of seasonally resolved stable isotope transects through fossil bivalves to reconstruct seasonality during the mid-Piacenzian warm period. The study is timely, well thought-out and very relevant for the readers of Climate of the Past.

I enjoyed reading about the combination of this extensive, high-quality dataset and would like to compliment the authors on bringing the data together to give the reader an overview of the seasonality reconstructions from different individuals (Figure 8). The images of the *Aequipecten* shells (Figure 1) and the overview of the stratigraphic context of the shells (Fig. 3, section 3) are also a very useful addition to the field! On reading through the manuscript, I did encounter some aspects of the discussion which may require a bit more attention, or with which I did not fully agree, and I wanted to highlight these below so the authors could consider them in their revision. These comments are meant to improve the discussion of the nice dataset that is presented, which by itself is already a very valuable contribution to the field and certainly merits publication.

Preservation

The authors acknowledge that no preservation screening was done on the shell material (lines 389-391). In most deep time (pre-Quaternary) sclerochronological studies, I would consider such an investigation essential to demonstrate the reliability of isotope records. Just a few trace element analyses to test against incorporation of Mn or Fe during diagenesis (see Brand and Veizer, 1980), XRD profiles to test original aragonite preservation in the aragonitic species and/or SEM images to demonstrate original shell structure preservation would lend more confidence to the interpretations in the manuscript. That said, the authors do cite evidence of good preservation of specimens from the same or time-equivalent deposits and I know from personal experience that the preservation of these shells from the Lillo formation show excellent preservation, so I would not consider the lack of preservation screening in this study to be a big obstacle to interpretation of the results.

**As noted in the comment, there is some published evidence of good preservation in the Pliocene sequence investigated (Valentine et al. 2011). It only extends to demonstration of original microstructures, but in both calcitic and aragonitic bivalve species, the investigated example of the former being an individual contributing isotopic data to the present study (as AO7; we will point out that it was this specific specimen that provided the evidence of good calcite preservation in Valentine et al. 2011). We took the view that the existence of pronounced δ$^{18}$O cyclicality, of a wavelength similar to that in modern examples of the same or similar species, was enough to confirm preservation of the original isotopic signature – a position explained in other studies (e.g. Johnson et al. 2017) but not stated explicitly here. Moon et al. (2021) have recently shown that dry heating at 200 °C can shift isotopic values but preserve cyclicality. Our shells, however, could only have experienced heating by a few degrees (in burial to a depth of little more than 100 m), so the existence of cyclicality can still be taken to indicate an original signature. I find it difficult to believe that cyclicality would be preserved in any (presumably fluid-based) process that introduced Fe or Mn into the shell after death, and high concentrations of Mn (as revealed by cathodoluminescence) can in any case result from incorporation during shell growth (e.g. Barbin et al. 1991; Soldati et al. 2010). I would therefore be reluctant to reject any of the present cyclical isotopic data on the basis of a high associated Mn concentration. It is worth noting here that a Pliocene aragonitic shell from the UK showing excellent mineralogical and microstructural preservation (*Cardites quamulosa ampla* 7 of Vignols et al. 2019) provided CL evidence of a high Mn content but yielded a δ$^{18}$O profile essentially the same as those from three other examples of the species in which Mn content was low from CL evidence. We will briefly discuss the evidence of good preservation provided by cyclicality in the revised version of the manuscript.**

**We have referred to AO7 as a specimen providing evidence of good preservation (L393). The work of Moon et al. (2021) is discussed in LL399–404. We accept that heating can alter absolute isotope values without disturbing cyclicality but point out that our material can only have been heated slightly in burial (by < 10 °C; LL390–392), in contrast to the intense heat (200 °C) involved in the experiments of Moon et al. (2021).**

Transfer functions

In the manuscript, the authors nicely discuss the effect of applying several different transfer functions for the d18O-temperature relationship and a range of potential d18O

values of the sea water on their d18O curves. Overall, I think this discussion is very honest and useful in showing the uncertainty on these d18O-based reconstructions, however I do not agree with the notion that the validity of transfer functions can be rejected or supported based on the data (e.g. lines 489-491; lines 679-684). In my opinion, the validity of proxy transfer functions like those for d18O can only be tested using modern carbonates precipitated at (approximately) known temperatures. Inferring the correctness of a transfer function based on the "fit" of fossil data with expected temperature outcomes runs the risk of circular reasoning. The discussion in lines 659-684, where outcomes of the d18O-temperature seasonality are compared with temperature reconstructions from ostracod and dinoflagellate assemblages is especially problematic, since the authors later (rightfully) argue that such assemblage-based reconstructions may be subject to bias (lines 926-929). My suggestion would be that the authors present the range of temperature seasonality outcomes they obtain from their fossil d18O data using various transfer functions and d18O values of the sea water as an uncertainty range. It is of course fine to discuss which outcomes fit better with previous reconstructions (which have their own uncertainty), but to conclude from these comparisons which transfer functions are best seems to push the interpretation a bit too far.

**I do not accept the charge of circular reasoning in the first half of the paragraph above and in the specific comment on lines 489–491 below. It results from a misinterpretation of the text in those lines. The δ¹⁸O data referenced (in Johnson et al. 2021b) are from a modern shell and the temperatures calculated using the various transfer functions were compared with directly measured values (just as recommended above for validation purposes). In so far as the δ¹⁸O data were mistakenly taken to be from Pliocene fossils and the temperature comparison was mistakenly taken to be with 'expected' Pliocene values, the text obviously needs some expansion/clarification. I will attend to this.**

**The expansion/clarification has been undertaken (LL504-505, 509).**

**While the above criticism is understandable (given the misinterpretation at its root), I am perplexed by the further criticism of comparisons between isotope- and assemblage-based estimates of Pliocene temperature in order to determine which of the former (based on different transfer functions) are more credible. It seems logical to give greater credence to estimates which are corroborated (even by 'uncertain' data) than to those which are uncorroborated.**

**No further response required.**

Statistics

In places where the uncertainty of the data is assessed (e.g. line 471-472) or comparisons between different records are made (e.g. line 570-574), the manuscript could benefit from more detailed statistical evaluation. For example, it would be more transparent if the measured values of the isotope standards are provided in a supplement and the actual mean value and standard deviation on these measurements is given in the text (line 471-472). In descriptions of the records, terms like "noise" (e.g. line 525) should be better defined and perhaps quantified. Statements like "substantially less variation" and "moderate positive covariation" (lines 570-574) should be backed up with statistical tests and quantification of uncertainty. Finally, I think the discussion would benefit from statistical evaluation of the seasonality outcomes and their uncertainty. The

comparison between temperature reconstructions, on which much of the discussion is based, is heavily dependent on the way in which seasonality is calculated and the degree by which differences between reconstructions are statistically significant. The authors discuss how their method for extracting seasonality from the extreme values of d18O records influences the outcome (e.g. section 5.1), but the study design using a large number of specimens (data in Fig. 8 and Table 2 and 3) should make it possible to calculate ranges and uncertainties for summer and winter temperatures, which can be used to test statistically if some species or combinations of assumed d18O of seawater and transfer functions are in agreement with previous temperature estimates (see paragraph above).

**As the seasonality outcomes are the principal 'result' of the paper I will deal first with the comments concerning these (second half of the above paragraph). We certainly needed to address uncertainties, and did so very thoroughly according to both referees. Full statistical comparisons with the data for winter and summer temperature now and (from other evidence) in the Pliocene are simply not possible because in neither case is there information on variation about the given values (indeed the data from Pliocene dinoflagellates are not in the form of specific temperatures but of temperature ranges: warm/cool temperate). I have, however, reflected on the comparisons that we did make and realized that a slightly different approach would be preferable. In Section 6.1.2 we started by comparing the difference between interval means for summer and winter temperature with the difference between the modern summer and winter seafloor temperatures at 53° N, 03° E (the site used in the validation exercise mentioned in response to the 'Transfer functions' comments). I think this was fair. However, we then went on to compare the largest single-year temperature ranges from individual shells with the seafloor seasonality figure for 53° N, 03° E, pointing out that some Pliocene ranges were higher than the latter figure. While this was worth recording, it does not mean that Pliocene seafloor seasonality was different from present. The data for 53° N, 03° E constitute a representative 'snapshot'; on the evidence of data from elsewhere in the southern North Sea (Lane and Prandle 1996) seasonal temperatures probably vary by ± 2 °C at this site. The 'high' Pliocene ranges can be accounted to this variation. The 'high' ranges, supplemented by a 3 °C stratification factor, were used in subsequent comparisons with the surface seasonality figure for 53° N, 03° E. This was not appropriate: the difference between interval means for summer and winter temperature should have been used, as in the comparisons of seafloor seasonality. While fairer, this approach makes very little difference to the figures for Pliocene surface seasonality, and revision of the data used will not require revision of the conclusions. It is worth noting here that in response to RC1 and RC2 the present seafloor temperature range at 53° N, 03° E has been indicated in Fig. 8, and the surface range has been included in a new plot (Fig. 9) showing individual and interval-mean summer temperatures incorporating the 3 °C ('minimum') stratification factor where appropriate - see the attached file. I think this visual representation of the Pliocene and modern data provides clear and convincing support to the argument in the text.**

**The consideration of individual data for seafloor temperature in relation to variation about the mean is at LL738–741 (with slightly modified text concerning interval mean data at LL719–724); the (new) consideration of interval mean data for sea surface temperature is at LL787–790 (with slightly modified text concerning individual data at LL790–797). Consideration of interval mean data has**

**entailed inclusion of an additional table (Table 4) with some modification of the content of the original Table 4 (now Table 5). Column headings have been slightly altered in Tables 2, 3, 5.**

Regarding the other statistical comments (first half of the above paragraph), I will be happy to include additional information to the extent that it is possible and worthwhile. For instance, the strength of covariation between $\delta^{13}C$ and $\delta^{18}O$ over parts of the ontogeny of individual specimens (lines 572–573) can be expressed in the form of a few $R^2$ values - information which can be added to the text without interrupting the overall flow (see also the response to RC1). By contrast, the weaker cyclicality in $\delta^{13}C$ compared to $\delta^{18}O$ would require substantial addition to the text if documented statistically – one would need to present and discuss test results relating to each $\delta^{18}O$ cycle in each specimen. I hardly think this level of statistical support is needed for a descriptive statement which is manifestly true from the evidence of Figs. 6 and 7. I will, however, modify the text concerning $\delta^{13}C$ to make it clear that the 'variation' referred to (line 570) is within ontogeny (i.e. over intervals comparable to those of $\delta^{18}O$ cycles) rather than over the whole of ontogeny. I can provide raw measurements of isotope standards for about half the analytical runs (conducted in my research over the last four years). For the other half (conducted in the PhD research of Annemarie Valentine from 2009 to 2013) I have only the measurements of samples, although analytical reproducibility was similar (Valentine et al. 2011). I personally think it is a bit excessive to provide raw measurements of standards (even as supplementary data) but will supply those from the more recent runs if it is considered worthwhile. Finally, noise is usually defined as 'unexplained variability'. Here we apply the term to low-amplitude, low-wavelength excursions from the higher-amplitude, higher-wavelength (cyclical) pattern of $\delta^{18}O$ variation, but also to a single relatively high-amplitude, low-wavelength excursion coincident with a similar excursion in $\delta^{13}C$. This usage is explained in the caption to Fig. 6d (the reference to Fig. 6c in line 525 will be corrected) and I think readers will readily recognise the common 'excursion' element – i.e. unexplained variability in the form of departures from a pattern.

**We have included $r^2$ values in relation to $\delta^{13}C/\delta^{18}O$ covariation (LL592, 594), clarified the text concerning $\delta^{13}C$ variation (LL588-589), elaborated on 'noise' (including mention of possible contamination; L542 and caption to Fig. 6) and corrected the reference to Fig. 6d from 6c (L542). I really do not think it is worth including incomplete raw data on standards in the supplementary information.**

Minor comments:

Line 128-129: In some species (e.g. *Crassostrea gigas*), shell sections in early ontogeny have been shown to by precipitated out of isotopic equilibrium (e.g. Huyghe et al., 2021), so this may not always be the best part of the shell to target for reconstructions.

**I was aware of the slightly earlier work of Huyghe et al. (2020) showing non-equilibrium (low) $\delta^{18}O$ values from the early ontogeny of *Crassostrea gigas* shells. Of the four species supplying $\delta^{18}O$ data for the present data, only *Aequipecten opercularis* was sampled over the first two years of growth. All the available evidence points to near-equilibrium isotopic incorporation in this phase of ontogeny, with year one providing the fullest and most accurate record of environmental temperature variation, as documented by Hickson et al. (1999) and Johnson et al. (2009, 2021b). These works, and those of Huyghe et al. (2020, 2021),**

could be discussed at this point, but it would perhaps over-elaborate the text. The works already cited provide adequate support for the existing general statement.

**We have referred to the results of Huyghe et al. (2020, 2021) from *Magallana* (= *Crassostrea*) *gigas* at LL129–131.**

Line 216-267: I really enjoyed reading this thorough review of the southern North Sea stratigraphy. I wonder if it would be beneficial to the reader to add rough paleo-depth curves to the sections in Fig. 3 to make the evolution of the paleoenvironment in these different areas easier to follow.

**The purpose of Fig. 3 is to show the stratigraphic relationships of units rather than their environment. Addition of palaeodepth curves would interfere with this objective. While fairly precise (albeit divergent) estimates are available for the sequence in northern Belgium, only very rough estimates (e.g. 'mainly above storm wavebase' for the Oosterhout Formation; Slupik et al. 2007) are available for the sequence in the south-west Netherlands.**

**No further response required.**

Line 405-407: Does this penetration of the resin into to shell affect the isotope analyses?

**The specimens concerned (*Glycymeris radiolyrata*) were investigated at Mainz, where I was asked not to present resin-contaminated material for analysis – I think because of calibration issues. While sampling therefore took place below the resin-contaminated zone (see Fig. 5), in a few cases boreholes did very slightly extend into it. There was no effect on the analytical results: $\delta^{18}O$ values were consistent with the pattern of those from completely uncontaminated samples before and after.**

**No further response required.**

Line 411: Figure 3 does not show the drilling of A. opercularis, but instead shows stratigraphy of the mPWP sections. Perhaps this should refer to Fig. 1? (although this figure also does not show the drill holes)

**The reference here is to the specimens illustrated in the cited papers by Hickson et al. – note the use of 'fig.' rather than 'Fig.', a common practice for referring to illustrations in other works.**

**No further response required.**

Line 480-482: The authors should briefly explain here why the global d18Osw values are rejected here.

**They give unreasonably low temperatures from *Aequipecten opercularis* – e.g. 0.1 °C and 1.6 °C (for water $\delta^{18}O$ values of –0.6 ‰ and –0.3 ‰, respectively) from AO6, a specimen from the *Atrina fragilis* bed of the Oorderen Member. Given the warm temperate summer temperature indicated by dinoflagellate evidence from this horizon, such extremely low winter temperatures are not credible. Any further consideration of them would reduce the credibility of the isotope-based temperatures as a whole. I will amend the text at this point accordingly.**

**Relevant text has been added at LL496–499.**

Line 489-491: See major comment above: The authors should explain why the Kim and O'Neil equation temperatures are "too low". I would be careful with this type of reasoning about transfer functions based on the "expected" temperature value.

**See the response to the major comment.**

**As indicated in the final response to the major comment, the text has been amended at LL504-505, 509.**

Line 546: Provide a number for "a great deal" to quantify the difference in growth rate.

**At the end of the sentence concerned I will add '(more than twice as fast as *A. islandica* and *P. rustica*, and three to five times faster than *G. radiolyrata*)'. This is in terms of the number of $\delta^{18}O$ cycles in a given height interval. Some would not accept this as a measure of growth rate (preferring a statistic relating to the whole of ontogeny; see discussion in Johnson et al. 2021a) so it is best not to say anything precise when all that is needed is general support for the statement 'a great deal faster'.**

**The text has been added at LL564, 565.**

Line 723: "overestimates" should be "overestimated"

**No – 'overestimates' is the correct word, referring to the fact that the calculated temperatures would be higher than the actual maximum temperatures.**

**No further response required.**

Line 750-752: See also major comment about the transfer function discussion: I wonder if this reasoning about the height of the stratification factor based on the temperature outcome and its comparison with modern temperatures is not sensitive to circular reasoning issues.

**In line 750 reference is made to Section 3, where independent (dinoflagellate) evidence of summer surface temperature is provided. I admit that this does not indicate warmer temperatures than now at all horizons but dinoflagellates give no indication that temperatures were ever cooler than now. I therefore think it is entirely reasonable to infer a summer surface temperature higher than the present value 600 km north of the study area, and thus a higher stratification factor.**

**We have amended the text to include mention of 'cool' dinoflagellate (dinocyst) biotas (LL782–784). However, we stand by the view that at the Pliocene locations in Belgium and Holland summer sea surface temperature was never cooler than now at a site in the central North Sea 600 km further north.**

Line 791-792: See comment above: I think one can almost never test the accuracy of proxy transfer functions (or the validity of d18Osw assumptions) based on their outcome on fossil data. This type of discussion requires independent evidence and/or modern calibration studies.

We don't pass any judgement on transfer functions in these lines, but have done so elsewhere (see the point above about the greater credibility of data that are corroborated).

**No further response required.**

Line 926-929: If the assumption of ecological uniformitarianism does not always hold (with which I agree), the authors should be careful with their conclusions from comparison of temperature reconstructions with the outcome of ostracod and dinoflagellate assemblage studies elsewhere in the discussion.

**We refer in lines 917–926 to evidence of niche change amongst bivalves. Until niche change is shown to be widespread and common I think we have to accept assemblage-based interpretations of palaeoenvironment founded on the ecology of modern representatives or close relatives of the species involved. There is certainly not much evidence as yet of niche change amongst ostracods and dinoflagellates.**

**No further response required.**

References

Brand, U. and Veizer, J.: Chemical diagenesis of a multicomponent carbonate system–1: Trace elements, 50, 1219–1236, 1980.

Huyghe, D., Daëron, M., de Rafelis, M., Blamart, D., Sébilo, M., Paulet, Y.-M., and Lartaud, F.: Clumped isotopes in modern marine bivalves, Geochimica et Cosmochimica Acta, https://doi.org/10.1016/j.gca.2021.09.019, 2021.

**Additional references**

**Barbin, V. Ramseyer, K., Debenay, J.P., Schein, E., Roux, M. and Decrouez, D., 1991. Cathodoluminescence of recent biogenic carbonates: environmental and ontogenetic fingerprint. Geological Magazine 128, 19–26. https://doi.org/10.1017/S001675680001801X.**

**Hickson, J.A., Johnson, A.L.A., Heaton, T.H.E. and Balson, P. S., 1999. The shell of the Queen Scallop *Aequipecten opercularis* (L.) as a promising tool for palaeoenvironmental reconstruction: evidence and reasons for equilibrium stable-isotope incorporation. Palaeogeography, Palaeoclimatology, Palaeoecology 154, 325–337. https://doi.org/10.1016/S0031-0182(99)00120-0.**

**Huyghe, D., Emmanuel, L., de Rafelis, M., Renard, M., Ropert, M., Labourdette, N. and Lartaud, F., 2020. Oxygen isotope disequilibrium in the juvenile portion of oyster shells biases seawater temperature reconstructions. Estuarine, Coastal and Shelf Science 240, article 106777. https://doi.org/10.1016/j.ecss.2020.106777.**

**Johnson, A.L.A., Hickson, J.A., Bird, A., Schöne, B.R., Balson, P.S., Heaton, T.H.E. and Williams, M., 2009. Comparative sclerochronology of modern and mid-Pliocene (c. 3.5 Ma) *Aequipecten opercularis* (Mollusca, Bivalvia): an insight into past and future climate change in the north-east Atlantic region.**

Palaeogeography, Palaeoclimatology, Palaeoecology 284, 164–179. https://doi.org/10.1016/j.palaeo.2009.0.022.

Johnson, A.L.A., Valentine, A., Leng, M.J., Sloane, H.J., Schöne, B.R. and Balson, P.S., 2017. Isotopic temperatures from the Early and Mid-Pliocene of the US Middle Atlantic Coastal Plain, and their implications for the cause of regional marine climate change. Palaios 32, 250–269. https://doi.org/10.2110/palo.2016.080.

Johnson, A.L.A., Harper, E.M., Clarke, A., Featherstone, A.C., Heywood, D.J., Richardson, K.E, Spink, J.O. and Thornton, L.A.H., 2021a. Growth rate, extinction and survival amongst ate Cenozoic bivalves of the North Atlantic. Historical Biology 33, 802-813. https://doi.org/10.1080/08912963.2019.1663839.

Johnson, A.L.A., Valentine, A.M., Schöne, B.R., Leng, M.J., Sloane, H.J. and Janeković, I., 2021b. Growth-increment characteristics and isotopic ($\delta^{18}$O) temperature record of sub-thermocline *Aequipecten opercularis* (Mollusca:Bivalvia): evidence from modern Adriatic forms and an application to early Pliocene examples from eastern England. Palaeogeography, Palaeoclimatology, Palaeoecology 561, article 110046. https://doi.org/10.1016/j.palaeo.2020.110046.

Lane, A. and Prandle, D., 1996. Inter-annual variability in the temperature of the North Sea. Continental Shelf Research 16, 1489–1507. https://doi.org/10.1016/0278-4343(96)00001-5.

Moon, L.R., Judd, E.J., Thomas, J. and Ivany, L.C., 2021. Out of the oven and into the fire: Unexpected preservation of the seasonal delta O-18 cycle following heating experiments on shell carbonate. Palaeogeography, Palaeoclimatology, Palaeoecology, article 110115. https://doi.org/10.1016/j.palaeo.2020.110115.

Slupik, A. A., Wesselingh, F. P., Janse, A. C. and Reumer, J. W. F., 2007. The stratigraphy of the Neogene–Quaternary succession in the southwest Netherlands from the Schelphoek borehole (42G4-11/42G0022)—a sequence stratigraphic approach. Netherlands Journal of Geoscience 86, 317–332. https://doi.org/10.1017/S0016774600023556.

Soldati, A.L., Goettlicher, J., Jacob, D.E., Vilas, V.V., 2010. Managanese speciation in *Diplodon chilensis patagonicus* shells: a XANES study. Journal of Synchrotron Radiation 17, 193–201. https://doi.org/10.1107/S090904950905465X.

Valentine, A., Johnson, A.L.A., Leng, M.J., Sloane, H.J. and Balson, P.S., 2011. Isotopic evidence of cool winter conditions in the mid-Piacenzian (Pliocene) of the southern North Sea Basin. Palaeogeography, Palaeoclimatology, Palaeoecology 309, 9–16. https://doi.org/10.1016/j.palaeo.2011.05.015.

Vignols, R.M., Valentine, A.M., Finlayson, A.G., Harper, E.M., Schöne, B.R., Leng, M.J., Sloane, H.J. and Johnson, A.L.A., 2019. Marine climate and hydrography of the Coralline Crag (early Pliocene, UK): isotopic evidence from 16 benthic invertebrate taxa, Chemical Geology 536, 62–83. https://doi.org/doi:10.1016/j.chemgeo.2018.05.034.

CC2 comments in normal typeface; response in **bold**)

I would like to thank the authors for their detailed replies to my comment on their work. I accept their solutions to the issues I raised, and concede that some of these were based on a misunderstanding (most notably my point about the rejection of d18O transfer functions based on fossil data). I appreciate the authors' clarification of this point.

I remain of the opinion that selecting the right transfer function for d18O-temperature conversion based on which temperatures fit better with (ostracod and dinoflagellate) assemblage results is prone to some uncertainty, especially if the authors want to make the point later in the paper that the nearest living relative approach underpinning these assemblage studies may not always be reliable. However, since I am not an expert on dinoflagellates and ostracods and since it would be hard to come up with an independent third line of evidence to (dis)prove their validity as temperature proxies, I think the authors' solution of presenting and discussing the data together and concluding what may be the most likely temperature seasonality given the evidence is a good one. I would just caution future authors in taking this discussion as evidence for the validity of one d18O-temperature relationship over another.

**We thank de Winter for his further comments. We certainly accept that our preference for one $\delta^{18}$O-temperature relationship over another because the outcomes fit with assemblage evidence of temperature in this case is no proof that the relationship is the best choice generally. More exacting $\delta^{18}$O work on modern shells (with accurate, location- and time-specific data on temperature and water $\delta^{18}$O) is one way to gain more certainty, as is $\Delta_{47}$ and/or biomineral-unit investigation of temperature alongside the $\delta^{18}$O approach in fossil shells. We suggest follow-up work using the last two techniques in this paper, and de Winter's expertise in $\Delta_{47}$ thermometry would certainly find a useful application here.**

**No further response required.**

---

## Author Response (AR2)

**Responses to editor's comments**

*I typed these responses in the 'reply' box associated with each comment in the PDF supplied. However, the replies would not save, so I have abstracted all the text and entered it below, with the line/figure number of the relevant 'sticky note' in the PDF.*

**L38**

**Editor:** A result of a stronger natural smoothing, since each measurment invcorporates a longer time interval than what will be integrated in the mean annual results based on the schlero-data?

**ALAJ:** Although individual shells give seasonal temperatures for individual years (and sequences of years) the seasonal values used for comparison with alkenone and $TEX_{86}$ temperatures are means of individual seasonal values over stratigraphic intervals. They are thus averages in much the same way as alkenone- and $TEX_{86}$-temperatures. The annual temperatures are similarly comparable, being the midpoint between (average) summer and winter values.

**L62**

**Editor:** suggested to be?

**ALAJ:** OK - I guess there is not yet a consensus. I have changed 'is now thought to be generally' to 'has now been suggested to be'.

**L78**

**Editor:** Is the reference to mid-point here from Dearing Crampton-Flood? If the winter signature of tex reflexts living depth rather than surface season, I will argue that its value will reflext annual mean independent of blooming season because temperatures are stable year round bellow the summer mixed layer (e.g. Jansen et al., 2008; Risebrobakken et al.,2011).

**ALAJ:** The notion of the mid-point being close to mean ASST comes from us. I have made this clear by various modifications of the text around here. Dearing Crampton-Flood et al. (2020, pp. 533-534) argue strongly that their $TEX_{86}$ data reflect winter surface temperature rather than (average) temperature at depth (the isoGDGT-2 / isoGDGT-3 ratio pointing towards a shallow water origin, the basin not being very deep anyway, and the main period of thaumarchaeotal blooms and associated isoGDGT production in the modern North Sea being in the winter months). I don't think this is the place for going into the evidence, but I have

drawn attention to its existence and made it clear that the TEX-86 and alkenone temperatures, are interpreted as winter and summer surface figures, respectively.

**L218**

**Editor:** specify what other issues you refer to here

**ALAJ:** They are issues about how to determine seasonal temperatures when growth rate is variable. Ivany and Judd (2022) favour a particular mathematical approach but it is not applicable to shells from sub-thermocline situations - i.e. the likely setting of some of those considered in this paper. I have explained this in the revised version.

**Fig. 6:**

**Editor:** I still consider the font size at the axises (numbers and axis names) to be too small and would recommend to increasing the size even more to make it readable. Now I need to blow it up to at least 200% to be able to read it. The same is the case for the legend and the microgrowth increment hight numbers. Compared to the font size used e.g. for (a), (b),... and AO3, AO4,... it is very small. It is correct that it will increase in the final version, but there is still room for improvement.

**ALAJ:** Increasing the size of the axis fonts reduces the size of the plot area in the Excel software used to create each part. To avoid this negative effect while still doing something to increase font sizes, I took the following actions: (1) in the originals of the 'AO' plots I increased the size of the smallest font used (for the values for increment-height variation in the green-bordered boxes) to that of the corresponding axis units; (2) in the ultimate figures (Figs 6, 7) I increased the size of all the parts about 4% by cropping the margins of each and then enlarging the remainder to fill the space created; (3) I increased the font size in the legend of Fig. 6 (which also applies to Fig. 7). I think these actions have addressed the legibility issues (compare the old and new versions in the 'tracked changes' file).

**L675**

**Editor:** Is something missing here? different what?

**ALAJ:** Yes, there are a few words missing here. Like the missing Fig. 8 and caption to Table 2 (see below), the missing words are present in the 'tracked changes' Word file from which I made the submitted PDF, so something 'funny' clearly happened in conversion. This may recur when I produce the 'new tracked changes' PDF but I will make sure that the 'clean' PDF is complete.

**L697**

**Editor:** Figure 8 is not added here - but I do see that its added in the version without track changes. And no Table caption is given for Table2 (but I do see it in the no track changes version)?

**ALAJ:** Both Fig. 8 and the caption to Table 2 were present in the 'tracked changes' Word file from which I made the submitted PDF. See the response to the comment above.

**L845**

**Editor:** Even when comparing to a annual mean from the schlero data presented here, the biomarkers are likely more smoothed since each measurement is likely to incorporate a longer time interval than the life time of one of the molluscs, so will the comparison be fair?

**ALAJ:** The comparison here is with mean seasonal temperatures from sclerochronology. I have changed 'interval mean temperatures' to 'interval-mean seasonal temperatures' to make this absolutely clear. See also the response to the first comment.

**L936**

**Editor:** This is a statement. What is the argumentation for why there is no reason to suppose?

**ALAJ:** The statement is based on independent evidence of environment and the latest interpretation of age. I have included a pointer to this information (provided in Section 3).